# EQUIVALENCE OF STATE EQUATIONS FROM DIFFERENT METHODS IN HIGH-DIMENSIONAL REGRESSION

## ABSTRACT

State equations (SEs) were firstly introduced in the approximate message passing (AMP) to describe the mean square error (MSE) in compressed sensing. Since then a set of state equations have appeared in studies of logistic regression, robust estimator and other high-dimensional statistics problems. Recently, a convex Gaussian min-max theorem (CGMT) approach was proposed to study high-dimensional statistic problems accompanying with another set of different state equations. This paper provides a uniform viewpoint on these methods and shows the equivalence of their reduction forms, which causes that the resulting SEs are essentially equivalent and can be converted into the same expression through parameter transformations. Combining these results, we show that these different state equations are derived from several equivalent reduction forms. We believe that this equivalence will shed light on discovering a deeper structure in high-dimensional statistics.

## 1 INTRODUCTION

Classical statistical methods often failed in the high-dimensional data where the number of features is larger than the number of observed samples. Studies in high dimensional data have attracted lots of attentions in past decades. A set of state equations (SEs) were first introduced in approximate message passing (AMP) algorithm in (Donoho et al., 2009) to precisely characterize the mean-square-error (MSE) and the phase transition phenomenon for true signal recovery in compressed sensing (CS). Since then, SEs, associated to certain AMP algorithm, have played indispensable role in various high-dimensional problems. For example, (Donoho et al., 2011) investigated the phase transition phenomenon and the precise MSE of LASSO estimator; (Donoho & Montanari, 2016) studied the variance of asymptotic distribution of M-estimator; (Huang, 2020) provided a precise characterization of min-max MSE of $l_1$ penalized robust M-estimator and the corresponding phase transition phenomenon.

Though the SEs were first introduced through certain AMP type algorithms, researchers meet them in a variety of models through different methods. For example, the SEs appeared in (El Karoui et al., 2013) when they performed the leaving-one-out (LOO) analysis of M-estimator in high dimensions. They showed that asymptotic normality, asymptotically unbiased property also hold as in the low dimension, nevertheless the variance of asymptotic distribution of M-estimators is higher. (Sur & Candès, 2019) employed the similar idea to analyze the properties of MLE in logistic regression where the SEs were used to show that (1) asymptotically unbiased property does not hold; (2) variance of asymptotic distribution increases; (3) likelihood ratio test is not distributed as chi-square. SEs also appeared in another line of researches where Thrompoulidies et al. performed analysis of a family of high dimensional problems through the Convex Gaussian min-max theorem (CGMT). More precisely, (Thrampoulidis et al., 2018) characterized the MSE precisely for general regularized M-estimator problem in high-dimensions; (Salehi et al., 2019) established the correlation and MSE of the resulting estimator of regularized logistic regression; (Deng et al., 2019) showed the changing trend of MSE with the growth of features in support vector machine and logistic regression.

Lastly, an insightful series of works (Barbier et al., 2019; Ricci-Tersenghi & Semerjian, 2009; Moore, 2014; Krzakala et al., 2016; Coja-Oghlan et al., 2018; Mézard & Parisi, 2003; Del Ferraro et al., 2014) have utilized the SEs (named as cavity method in statistical physics) as a ubiquitous tool when they studied the high dimensional statistical problem through the perspective of statisti-

cal physics. Importantly, this tool has exhibited as a powerful weapon in applications of a lot of fields(Mezard & Montanari, 2009; Obuchi & Kabashima, 2016; Vuffray, 2014; Lesieur et al., 2015; 2016).

Though many papers have explicitly written down the corresponding state equations, none of them have shown that these sets of state equations are compatible. To the best of our knowledge, only (Deng et al., 2019) mentioned there is another set of state equation but without any comparison.

Although SEs were proved to be important in high dimensional problems, it is awkward that for one specific problem, the resulting SEs from AMP, CGMT and LOO are different. To be more clear, let us take a look at logistic regression. The SEs derived from CGMT (20) in (Deng et al., 2019) are obviously different from the SEs derived from LOO (19) in (Sur & Candès, 2019). This is annoying, since the asymptotic performance for a specific high-dimensional problem should be unique no matter which method was used.

Therefore, we are interested in the following questions:

*Are SEs derived from different methods all equivalent in some sense? If so, from what viewpoint these methods are equivalent and are there more inner equivalence?*

Among them, as the most direct, accessible, basic tool, equivalence of SEs is the basis of equivalence of methods and more inner equivalence.

**Our contributions.** We successfully show that for various high dimensional problem, the different sets of SEs derived through different methods are actually equivalent to each other. More precisely, we construct the equivalence between different sets of SEs through explicit parameter transforms for LASSO, M-estimator and logistic regression. These transformations are inspired by the statistical meanings of certain quantities appeared in the SEs. Moreover, we also provide a heuristic explanation on the relation between the different methods: AMP, CGMT and LOO. To the best of our knowledge, this is the first work to clearly clarify the equivalence among SEs derived from different methods and try to establish the equivalence of different methods.

**Outlines.** In section 2, we show that the SEs for M-estimator from AMP, LOO and CGMT are equivalent to each other. In section 3.1, we show the equivalence of SEs derived from AMP and CGMT for another example and explain the essential reasons behind this equivalence. In Section 3.2, we illustrate the similar work regarding the equivalence between CGMT and LOO. Section 4 provides some discussions and future directions. Most proofs are deferred to the appendix.

**Notations.** Let $\mathcal{N}(\mathbf{0}, \boldsymbol{I}_d), \mathcal{N}(\mathbf{0}, 1)$ denote the $d$-dimensional standard Gaussian distribution and 1-dimensional standard Gaussian distribution respectively. For a vector $\boldsymbol{x}$, we denote $\|\boldsymbol{x}\|_p$ as the $l_p$ norm of $\boldsymbol{x}$. For an integer $n$ we denote $[n]$ as $\{1, \cdots, n\}$. We abbreviate independent and identically distributed to i.i.d.. For a function $f : \mathbb{R} \mapsto \mathbb{R}$, variable $x \in \mathbb{R}$ and $t > 0$, we denote the Moreau envelope associated with $f$ as

$$M_f(x; t) := \min_{z \in \mathbb{R}} f(z) + \frac{1}{2t}(x - z)^2 \tag{1}$$

and the proximal operator, which is the solution of this minimization as

$$Prox_f(x; t) := \arg\min_{z \in \mathbb{R}} f(z) + \frac{1}{2t}(x - z)^2. \tag{2}$$

For multi-dimensional case $\boldsymbol{x} = (x_1, \cdots, x_d)^T \in \mathbb{R}^d$, Moreau envelope and proximal operator are applied element-wisely: $M_f(\boldsymbol{x}; t) := (M_f(x_i; t)) \in \mathbb{R}^d$ and $Prox_f(\boldsymbol{x}; t) := (Prox_f(x_i; t)) \in \mathbb{R}^d$.

## 2    AN ILLUSTRATIVE EXAMPLE

Suppose that $\mathbf{x}_i \overset{\text{i.i.d.}}{\sim} \mathcal{N}(0, \frac{1}{d}\boldsymbol{I}_d)$ and $\mathbf{y}_i \in \mathbb{R}$ satisfying that

$$\mathbf{y}_i = \mathbf{x}_i^T \boldsymbol{\beta}^* + \epsilon_i, \quad \text{for } i \in [n] \tag{3}$$

where $\epsilon_i$ are drawn i.i.d. from distribution $P_\epsilon$ with mean 0 and variance $\sigma_*^2$. We assume that the entries $\beta_i^*$ of $\boldsymbol{\beta}^*$ are independently distributed as $\Pi$ which has finite second moment $r_*^2 = \mathbb{E}_{\beta \sim \Pi}\beta^2$.

Let $\rho$ be a non-negative convex function. We are interested in the the Mean-squared-error (MSE) performance $\lim_{n,p\to\infty} \frac{1}{n}\|\boldsymbol{\beta} - \boldsymbol{\beta}^*\|^2$ of the M-estimator:

$$\hat{\boldsymbol{\beta}} = \arg\min_{\boldsymbol{\beta}} \sum_{i=1}^{n} \rho(\mathbf{y}_i - \mathbf{x}_i^T\boldsymbol{\beta}) \tag{4}$$

when both $n$ and $d$ go to infinity satisfying that $\lim_{n,d\to\infty} \frac{d}{n} = \kappa_* \in (0,\infty)$.

This problem first studied by (El Karoui et al., 2013) where they showed that the MSE of $\hat{\beta}$ can be characterized by a set of SEs. More precisely, they proved the following proposition.

**Proposition 2.1.** *(El Karoui et al., 2013) Given ratio $\kappa_* < 1$. Consider the following system of nonlinear equations (SEs) regarding $(\tau_1, \gamma_1)$ :*

$$1 - \kappa_* = \mathbb{E}[\frac{\partial Prox_\rho}{\partial x}(W_1 + \tau_1 Z_1; \lambda_1)]$$
$$\kappa_* \tau_1^2 := \mathbb{E}[W_1 + \tau_1 Z_1 - Prox_\rho(W_1 + \tau_1 Z_1; \lambda_1)]^2 \tag{5}$$

*where $W_1 \sim P_\epsilon, Z_1 \sim \mathcal{N}(0,1)$ is independent of $W_1$. If this system of nonlinear equations possesses a unique solution $(\bar{\tau}_1, \bar{\lambda}_1)$, then the $\bar{\tau}_1$ is exactly the MSE of $\hat{\boldsymbol{\beta}}$ appeared in (4).*

The $M$-estimator was also studied by (Donoho & Montanari, 2016) where they proved the following proposition.

**Proposition 2.2.** *(Donoho & Montanari, 2016) Given ratio $\kappa_* < 1$. Consider the following system of nonlinear equations (SEs) regarding $(\tau_2, \gamma_2)$ :*

$$\tau_2^2 = \frac{1}{\kappa_*} \lambda_2^2 \mathbb{E}[\frac{\partial M_\rho}{\partial x}(W_2 + \tau_2 Z_2; \lambda_2)]^2$$
$$\kappa_* = \lambda_2 \mathbb{E}[\frac{\partial^2 M_\rho}{\partial x^2}(W_2 + \tau_2 Z_2; \lambda_2)] \tag{6}$$

*where $W_2 \sim P_\epsilon, Z_2 \sim \mathcal{N}(0,1)$ is independent of $W_2$. If this system of nonlinear equations possesses a unique solution $(\bar{\tau}_2, \bar{\lambda}_2)$, then the $\bar{\tau}_2$ is exactly the MSE of $\hat{\boldsymbol{\beta}}$ appeared in (4).*

Moreover, inspired by the work (Thrampoulidis et al., 2014), we employ the CGMT techniques to study the $M$-estimator and show that the asymptotic MSE can be characterized by the the following SEs. To avoid unnecessary digression, we defer the detailed proof to the appendix A.

**Proposition 2.3.** *Given ratio $\kappa_* < 1$. Consider the following system of nonlinear equations (SEs) regarding $(\tau_3, \alpha, \mu)$ :*

$$0 = \frac{\alpha}{2} - \tau_3\sqrt{\kappa_*} - \frac{\alpha}{\mu^2}\mathbb{E}[\frac{\partial M_\rho}{\partial t}(W_3 + \tau_3 Z_3; \alpha/\mu)]$$
$$0 = -\mu\sqrt{\kappa_*} + \mathbb{E}[Z_3 \frac{\partial M_\rho}{\partial x}(W_3 + \tau_3 Z_3; \alpha/\mu)] \tag{7}$$
$$0 = \frac{\mu}{2} + \frac{1}{\mu}\mathbb{E}[\frac{\partial M_\rho}{\partial t}(W_3 + \tau_3 Z_3; \alpha/\mu)]$$

*where $W_3 \sim P_\epsilon, Z_3 \sim \mathcal{N}(0,1)$ is independent of $W_3$. If this system of nonlinear equations possesses a unique solution $(\bar{\tau}_3, \bar{\alpha}, \bar{\mu})$, then the $\bar{\tau}_3$ is exactly the MSE of $\hat{\boldsymbol{\beta}}$ appeared in (4).*

On the one hand, these three sets of SEs are different at the first glance. On the other hand, since they are all supposed to describe the MSE of the $M$-estimators in high dimension, there shall be some relation between these three sets of equations. A striking fact is that we can actually show that all these three set of SEs are equivalent to each other. More precisely, we have the following theorem.

**Theorem 1.** *For M-estimator(4), the SEs derived from AMP (6), LOO (5) and CGMT (7) are equivalent. Specifically, (6) can be converted into the same form as (5) after the following parameter transformations:*

$$\tau_1 = \tau_2, \quad \lambda_1 = \lambda_2. \tag{8}$$

*(6) can be converted into the same form as (7) after the following parameter transformations:*

$$\tau_1 = \tau_3, \quad \lambda_1 = \frac{\alpha}{\mu}. \tag{9}$$

The equivalence of these three sets of SEs seems straightforward, however, it suggests us that all the three procedures: AMP, CGMT and LOO might be deeply entangled in some sense. This will be investigated in this manuscript.

The proof of this theorem is deferred to the appendix B.

## 3  GENERAL RESULTS

In this section, we show that the aforementioned equivalence between different sets of SEs holds not only for M-estimator, but also for Lasso and logistic regression in high dimensions.

### 3.1  EQUIVALENCE BETWEEN THE SES DERIVED FROM CGMT AND AMP

Let us consider the following optimization

$$\min_{\boldsymbol{\beta}} \frac{1}{2}\|\mathbf{y} - \mathbf{X}\boldsymbol{\beta}\|_2^2 + \lambda_*\|\boldsymbol{\beta}\|_1 \tag{10}$$

where $y_i = \mathbf{x}_i^\tau \boldsymbol{\beta}^* + \epsilon_i$ with $\mathbf{x}_i \overset{\text{i.i.d.}}{\sim} \mathcal{N}(0, \frac{1}{n}\boldsymbol{I}_d)$, $r_* := \lim_{n,p\to\infty} \frac{\|\boldsymbol{\beta}^*\|}{\sqrt{n}}$ and $\lambda_* \geq 0$ is the regularized parameter, $\epsilon_i$ are drawn i.i.d. from distribution $P_\epsilon$ with mean 0 and variance $\sigma_*^2$.

We are interested in the the Mean-squared-error(MSE) performance $\lim_{n,p\to\infty} \frac{1}{n}\|\boldsymbol{\beta} - \boldsymbol{\beta}^*\|^2$ of the LASSO. (Donoho et al., 2011), (Mousavi et al., 2018), (Bayati & Montanari, 2011; Miolane & Montanari, 2018; Javanmard & Montanari, 2018) have utilized the AMP to study the asymptotic performance of the Lasso estimator. For our purpose, we briefly recall the results in (Mousavi et al., 2018) below.

**Proposition 3.1.** *(Mousavi et al., 2018) Given noise scale $\sigma_*^2$ and ratio $\kappa_*$, consider the following system of nonlinear equations (SEs) regarding $(\tau_1, \gamma_1)$ :*

$$\begin{aligned} \tau_1^2 &= \sigma_*^2 + \kappa_* \mathbb{E}[\eta(\beta_1 + \tau_1 Z_1; \lambda_* + \gamma_1) - \beta_1]^2 \\ \gamma_1 &= \kappa_*(\gamma_1 + \lambda_*)\mathbb{E}[\eta'(\beta_1 + \tau_1 Z_1; \lambda_* + \gamma_1)] \end{aligned} \tag{11}$$

*where $Z_1 \sim \mathcal{N}(0, 1)$ is a standard normal variable, $\beta_1 \sim \Pi$ is independent of $Z_1$, $\eta(\cdot; \cdot)$ is the soft threshold function:*

$$\eta(x; t) := sign(x)(|x| - t)_+,$$

*$x_+$ means $\max\{x, 0\}$ and*

$$sign(x) := \begin{cases} 1 & \text{if } x > 0 \\ 0 & \text{if } x = 0. \\ -1 & \text{if } x < 0 \end{cases}$$

*If this system of nonlinear equations possesses a unique solution $(\bar{\tau}_1, \bar{\lambda}_1)$, then the $\bar{\tau}_1$ is exactly the MSE of $\hat{\boldsymbol{\beta}}$ appeared in (10).*

Inspired by the sequence of work (Thrampoulidis et al., 2014; 2015; 2018; Salehi et al., 2019), we apply the CGMT to study the asymptotic performance of the Lasso estimator appeared in (10) and find that it is characterized by the following set of SEs.

**Proposition 3.2.** *Given noise scale $\sigma_*^2$, signal strength $r_*^2$ in model (3) and ratio $\kappa_*$, Consider the following system of nonlinear equations (SEs) regarding $(\alpha, \sigma, \tau_2, \theta, \lambda, \gamma_2)$:*

$$
\begin{aligned}
0 &= -\frac{\alpha}{\sigma\tau_2} + \theta - 1 + \frac{\alpha + \lambda}{\lambda + 1} \\
0 &= -\frac{1}{2\tau_2} + \frac{r_*^2\kappa_*\alpha^2}{2\sigma^2\tau_2} - \frac{\tau_2\kappa_*}{2}\mathbb{E}[(Prox_{\tilde{f}}(\gamma_2 Z_2 + \theta\beta_2; \lambda_*))^2] + \frac{\sigma}{\lambda + 1} \\
0 &= \gamma_2^2 - r_*^2\kappa_* - \sigma_*^2 + \frac{2[(\alpha + \lambda)r_*^2\kappa_* + \lambda\sigma_*^2]}{\lambda + 1} - \frac{(\alpha + \lambda)^2 r_*^2\kappa_* + \sigma^2 + \lambda^2\sigma_*^2}{(\lambda + 1)^2} \\
0 &= r_*^2\kappa_*\alpha - \sigma\tau_2\kappa_*\mathbb{E}[\beta_2 Prox_{\tilde{f}}(\gamma_2 Z_2 + \theta\beta_2; \lambda_*)] \\
\lambda &= \sigma\tau_2\kappa_*\mathbb{E}[\frac{\partial Prox_{\tilde{f}}(\gamma_2 Z_2 + \theta\beta_2; \lambda_*)}{\partial x}] \\
0 &= \frac{\sigma}{2\tau_2^2} + \frac{r_*^2\kappa_*\alpha^2}{2\sigma\tau_2^2} - \frac{\sigma\kappa_*}{2}\mathbb{E}[(Prox_{\tilde{f}}(\gamma_2 Z_2 + \theta\beta_2; \lambda_*))^2]
\end{aligned}
\tag{12}
$$

*where $Z_2 \sim \mathcal{N}(0, 1)$ is a standard normal variable, $\beta_2 \sim \Pi$ is independent of $Z_2$, $\tilde{f}(x) := |x|$.*

*If this system of nonlinear equations possesses a unique solution $(\bar{\alpha}, \bar{\sigma}, \bar{\tau}_2, \bar{\theta}, \bar{\lambda}, \bar{\gamma}_2)$, then the $\frac{\bar{\lambda}_2}{\bar{\theta}}$ is exactly the MSE of $\hat{\beta}$ appeared in (10)*

The detailed proof is deferred until the Appendix C. The following proposition illustrate the equivalence between these two sets of SEs.

**Theorem 2.** *The SEs of LASSO derived from AMP (11) are equivalent to the SEs derived from CGMT (12). Specifically, (12) can be converted into the same form as (11) after the following parameter transformations:*

$$
\tau_1 = \frac{\gamma_2}{\theta}, \quad \gamma_1 = \frac{\lambda_*}{\theta} - \lambda_*.
\tag{13}
$$

The detailed proof is deferred until Appendix C.1.

We provided a heuristic explanation on the equivalence of the SEs derived from AMP and CGMT. For the sake of the self-contentment, we briefly review the procedures of how to derive SEs from AMP and CGMT respectively.

**Deriving SEs from AMP.** The derivation of SE from AMP can be divided into two stages:

(1) Constructing an iterative algorithm

   1) AMP first transform initial optimization problem into pursuing a Bayesian posterior distribution where objective function is transformed into a probability distribution.
   2) Based on the corresponding factor graph of this distribution, it invokes the message passing(MP) algorithm to compute the Bayesian posterior distribution.
   3) The MP is then further approximated by some large system limit, large $\beta$ limit and the approximation of iteration.

(2) The asymptotic behavior of AMP is then characterized by the state evolution equations/SEs.

**Deriving SEs from CGMT.** The derivation of SE from CGMT can be divided into four steps.

   1) The initial optimization problem is transformed into a min-max form, which is called the primary optimization (PO) problem.
   2) CGMT perform a dimensionality reduction on PO and obtain the auxiliary optimization (AO) problem
   3) AO is further simplified to an optimization problem only depending on several scalar variables, which is called scalar optimization (SO) problem.

4) SEs are derived by finding first-order optimality conditions of the asymptotic version of SO.

**Remark 3.1.** *We find that AO can be viewed as a relaxation of PO in the sense that the feasible region of AO is larger than that of PO. Concrete examples, such as M-estimator, Logistic regression, Support vector machine and so on, are deferred to the appendix. We believe that this relaxation can help us understand the equivalence between the resulting SEs from AMP and CGMT respectively.*

We now present a uniform viewpoint on AMP and CGMT: 1) Constructing the AMP corresponds to the first step of CGMT in LASSO, which suggests that the iteration of AMP is actually equivalent to the process of solving PO. 2) Deriving the SEs from AMP corresponds to the last three steps of CGMT. Both of them aim to deriving SEs and characterizing asymptotic performance by approximating the initial optimization problem. We proved the first statement in Proposition 3.3.

**Proposition 3.3.** *(Rangan et al., 2016) For LASSO, the fixed point of AMP is just the solution of first-order optimality conditions of PO in CGMT.*

*Proof.* For CGMT, by introducing $\boldsymbol{u}$ to constrain $\boldsymbol{u} = \mathbf{X}\boldsymbol{\beta}$ and Lagrange vector $\boldsymbol{v}$, the corresponding PO can be written as:

$$\min_{\boldsymbol{\beta},\boldsymbol{u}} \max_{\boldsymbol{v}} \frac{1}{2}\|\boldsymbol{u}\|_2^2 - \mathbf{y}^T\boldsymbol{u} + \frac{1}{2}\|\mathbf{y}\|_2^2 + \lambda_*\|\boldsymbol{\beta}\|_1 + \boldsymbol{v}^T(\mathbf{X}\boldsymbol{\beta} - \boldsymbol{u}).$$

Consider the first-order optimality conditions of PO:

$$\begin{cases} 0 = \lambda_* \text{sign}(\boldsymbol{\beta}) + \mathbf{X}^T\boldsymbol{v} \\ 0 = \boldsymbol{u} - \mathbf{y} - \boldsymbol{v} \\ 0 = \boldsymbol{u} - \mathbf{X}\boldsymbol{\beta}. \end{cases} \tag{14}$$

Comparing above formulas in (14) leads to

$$\lambda_* \text{sign}(\boldsymbol{\beta}) + \mathbf{X}^T(\mathbf{X}\boldsymbol{\beta} - \mathbf{y}) = 0.$$

For AMP algorithm, the iteration of LASSO is

$$\boldsymbol{\beta}^{t+1} = \eta(\boldsymbol{\beta}^t + \mathbf{X}^T\boldsymbol{z}^t; \lambda_* + \gamma^t)$$

$$\boldsymbol{z}^t = \mathbf{y} - \mathbf{X}\boldsymbol{\beta}^t + \kappa_*\boldsymbol{z}^{t-1}\langle\frac{\partial}{\partial x}\eta(\boldsymbol{\beta}^{t-1} + \mathbf{X}^T\boldsymbol{z}^{t-1}; \lambda_* + \gamma^{t-1})\rangle$$

$$\gamma^t = \kappa_*(\lambda_* + \gamma^{t-1})\langle\frac{\partial}{\partial x}\eta(\boldsymbol{\beta}^{t-1} + \mathbf{X}^T\boldsymbol{z}^{t-1}; \lambda_* + \gamma^{t-1})\rangle$$

where $\frac{\partial}{\partial x}$ acts component-wisely. For some vector $\boldsymbol{x}$, $\langle\boldsymbol{x}\rangle := \sum_{i=1}^d x_i$ denotes the entry-sum of $\boldsymbol{x}$.
The fixed point $(\boldsymbol{\beta}^\infty, \boldsymbol{z}^\infty, \gamma^\infty)$ satisfy the following equations:

$$0 = (\lambda_* + \gamma)\text{sign}(\boldsymbol{\beta}) + \boldsymbol{\beta} - (\boldsymbol{\beta} + \mathbf{X}^T\boldsymbol{z}) \tag{15a}$$

$$\boldsymbol{z} = \mathbf{y} - \mathbf{X}\boldsymbol{\beta} + \kappa_*\boldsymbol{z} \cdot c \tag{15b}$$

$$\gamma = \kappa_*(\lambda_* + \gamma)c \tag{15c}$$

where $c = c(\boldsymbol{\beta}, \boldsymbol{z}, \gamma) = \langle\frac{\partial}{\partial x}\eta(\boldsymbol{\beta} + \mathbf{X}^T\boldsymbol{z}; \lambda_* + \gamma)\rangle$ and (15a) is given by the following property about the soft thresholding function:

$$t \cdot \text{sign}(z) + z - x = 0$$

for $z = \eta(x; t)$ and some scalar $x$.

Simplifying (15b) and (15c) leads to:

$$\boldsymbol{z} = \frac{\mathbf{y} - \mathbf{X}\boldsymbol{\beta}}{1 - \kappa_*c}$$

$$\gamma = \frac{\kappa_*c\lambda_*}{1 - \kappa_*c}. \tag{16}$$

Comparing (16) with (15a) gives,

$$\frac{\lambda_*}{1 - \kappa_* c} \text{sign}(\boldsymbol{\beta}) - \frac{1}{1 - \kappa_* c} \mathbf{X}^T (\mathbf{y} - \mathbf{X}\boldsymbol{\beta}) = 0.$$

which finishes the proof. $\qquad\square$

**Remark 3.2.** *It needs to be discussed component-wisely according to whether each entry of the optimal $\boldsymbol{\beta}$ is 0 or not. The above proof holds for the entries that $\beta_i \neq 0$. For $i$ such that $\beta_i = 0$, the optimality from PO gives $-\lambda_* + (\mathbf{X}^T(\mathbf{X}\beta - \mathbf{y}))_i < 0$ and $\lambda_* + (\mathbf{X}^T(\mathbf{X}\beta - \mathbf{y}))_i > 0$. This is equivalent to $|(\mathbf{X}^T(\mathbf{X}\beta - \mathbf{y}))_i| \leq \lambda_*$, where $(\mathbf{X}^T(\mathbf{X}\beta - \mathbf{y}))_i$ denote the $i$-th entry of $\mathbf{X}^T(\mathbf{X}\beta - \mathbf{y})$. This is still equivalent to the fixed-point condition in AMP. Hence the equivalence holds for all entries of $\boldsymbol{\beta}$.*

### 3.2 EQUIVALENCE BETWEEN THE SEs DERIVED FROM CGMT AND LOO

Suppose that $\mathbf{x}_i \overset{\text{i.i.d.}}{\sim} \mathcal{N}(0, \frac{1}{d}\boldsymbol{I}_d)$ and $y_i \in \{-1, 1\}$ drawn from logistic model:

$$\mathbb{P}(\mathbf{y}_i = 1|\mathbf{x}_i) = \rho'(\mathbf{x}_i^T \boldsymbol{\beta}^*), \quad \text{for } i \in [n] \qquad (17)$$

where $\rho(t) = \log(1 + e^t)$. Each entry of $\boldsymbol{\beta}$ is independently distributed as $\Pi$ which has finite second moment $r_*^2 = \mathbb{E}_{\beta \sim \Pi} \beta^2$.

We are interested in the following optimization problem:

$$\hat{\boldsymbol{\beta}} = \arg \min_{\boldsymbol{\beta}} \frac{1}{n} \sum_{i=1}^n \ell(\mathbf{y}_i \mathbf{x}_i^T \boldsymbol{\beta}) \qquad (18)$$

where $\ell(t) := \log(1 + e^{-t})$. When the $\hat{\boldsymbol{\beta}}$ exists, we are interested in the the Mean-squared-error(MSE) performance $\lim_{n,p \to \infty} \frac{1}{n} \|\boldsymbol{\beta} - \boldsymbol{\beta}^*\|^2$ of the Logistic regression.

Logistic regression in high dimensions have been studied recently by (Candès & Sur, 2020; Mousavi et al., 2018), (Deng et al., 2019). The asymptotic MSE of $\hat{\beta}$ was characterized by the following two propositions.

**Proposition 3.4.** *(Sur & Candès, 2019) Given signal strength $r_*^2$ in logistic model (17) and ratio $\kappa_*$, Consider the following system of nonlinear equations (SEs) regarding $(\lambda_1, \alpha_1, \sigma)$:*

$$\alpha_1^2 = \frac{1}{\kappa_*^2} \mathbb{E}[2\rho'(Q_1) \left(\lambda_1 \rho'(Prox_\rho(Q_2; \lambda_1))\right)^2]$$

$$0 = \mathbb{E}[\rho'(Q_1)Q_1 \lambda_1 \rho'(Prox_\rho(Q_2; \lambda_1))] \qquad (19)$$

$$1 - \kappa_* = \mathbb{E}[\frac{2\rho'(Q_1)}{1 + \lambda_1 \rho''(Prox_\rho(Q_2; \lambda_1))}]$$

*where*

$$(Q_1, Q_2) \sim \mathcal{N}\left(\mathbf{0}; \begin{bmatrix} r_*^2 & -\sigma r_*^2 \\ -\sigma r_*^2 & \sigma^2 r_*^2 + \alpha_1^2 \kappa_* \end{bmatrix}\right)$$

*and $\rho(t) := \log(1 + e^t)$.*

*If this system of nonlinear equations possesses a unique solution $(\bar{\lambda}_1, \bar{\alpha}_1, \bar{\sigma})$, then the MSE of $\hat{\boldsymbol{\beta}}$ appeared in (18) is $[(\bar{\sigma} - 1)\mathbb{E}_{\beta \sim \Pi}\beta]^2 + \bar{\alpha}^2$.*

**Remark 3.3.** *In (Sur & Candès, 2019), it is assumed that $X_{i,j} \sim \mathcal{N}(0, \frac{1}{n}I_d)$ and $r_*^2 = \kappa_* \mathbb{E}_{\beta \sim \Pi}\beta^2$, which is slightly different from the setting in this paper. However, this difference only leads to a constant change related to $\kappa_*$ in the final parameter transformations (21) and does not affect the equivalence of these two set of SE.*

**Proposition 3.5.** *(Deng et al., 2019) Given signal strength $r_*^2$ in logistic model (17) and ratio $\kappa_*$, Consider the following system of nonlinear equations (SEs) regarding $(\lambda_2, \alpha_2, \mu)$:*

$$0 = \mathbb{E}[V\ell'(Prox_\ell(\alpha_2 Z + \mu V; \lambda_2))]$$

$$\alpha_2^2 \kappa_* = \lambda_2^2 \mathbb{E}[(\ell'(Prox_\ell(\alpha_2 Z + \mu V; \lambda_2)))^2] \qquad (20)$$

$$\kappa_* = \lambda_2 \mathbb{E}[\frac{\ell''(Prox_\ell(\alpha_2 Z + \mu V; \lambda_2))}{1 + \lambda \ell''(Prox_\ell(\alpha_2 Z + \mu V; \lambda_2))}]$$

where $Z \sim \mathcal{N}(0,1)$, $V = Z_1 Y_{r_*}$, in which $Z_1 \sim \mathcal{N}(0,1)$ is independent of $Z$ and $Y_{r_*} \sim Ber(\rho'(r_* Z_1))$. $Ber(p)$ denotes the Bernoulli distribution with probability $p$ for the value $+1$ and probability $1-p$ for the value $-1$. If this system of nonlinear equations possesses a unique solution $(\bar{\lambda}_2, \bar{\alpha}_2, \bar{\mu})$, then the MSE of $\hat{\boldsymbol{\beta}}$ appeared in (18) is $[(\frac{\bar{\alpha}_2}{\sqrt{\kappa_*}} - 1)\mathbb{E}_{\beta \sim \Pi}\beta]^2 + (\frac{\bar{\mu}}{r_*})^2$.

As before, we can show that (19) and (20) are equivalent.

**Theorem 3.** *For logistic regression (18), the SEs derived from LOO (19) and CGMT (20) are equivalent. Specifically, (19) can be converted into the same form as (20) after the following parameter transformations:*

$$\alpha_1 = \frac{\alpha_2}{\sqrt{\kappa_*}}, \quad \sigma = \frac{\mu}{r_*}, \quad \lambda_1 = \lambda_2. \tag{21}$$

The proof of this theorem is deferred to appendix E

For the sake of self-contentment, we briefly review the procedure on deriving SEs from LOO.

**Deriving SEs from LOO** The derivation can be divided into 4 steps.

1) First, for the original optimization problem, LOO considers first-order conditions of three cases: a) keeping all observations and predictors, corresponding solution is denoted as $\hat{\beta}$, b) leaving one predictor, corresponding solution is denoted as $\hat{\beta}_{(-j)}$ and c) leaving one predictor and one observation, corresponding solution is denoted as $\hat{\beta}_{(-i),(-j)}$

2) Two properties are derived from comparing three version of first-order conditions: a) The $i$-th fitted value $X_i\hat{\beta}$ has an asymptotic expression composed of two independent random vectors $X_{i,(-j)}$ and $\hat{\beta}_{(-i),(-j)}$. b) Each coordinate $\hat{\beta}_j$ can be written as a sum of $n$ random variables which are asymptotically independent.

3) Using above two properties , $\hat{\beta}_j$ has the same distribution as a combination of several scalar variables when $n, p \to \infty$. Hence every statistic of $\hat{\beta}$ (such as expectation, variance and first order condition of optimization) can be expressed by these scalar variables, from which the SEs of $\hat{\beta}$ are derived.

Briefly reviewing the procedures of LOO approach, we find that the sample matrix $X$ (which is a $\mathbb{R}^{n \times p}$ Gaussian matrix)is decomposed into two independent Gaussian vectors through some special techniques in both LOO and CGMT, which allows the law of large numbers to simplify the first-order equations into scalar equations. This may help us understand the equivalence between CGMT and LOO. The more intrinsic equivalence of these two methods is still under investigation.

## 4    DISCUSSION AND FUTURE DIRECTIONS

In this paper, we first showed that for the high dimensional $M$-estimator, the three sets of SEs derived from AMP, CGMT and LOO are equivalent. We then further show that this equivalence actually appears in various high dimensional problems. This strongly suggests us that there should be a deep relation between these three approaches.

Though AMP, CGMT and LOO are different at the first glance, we find that they all can be treated as approximations of the same first order optimality conditions. To be more precise, LOO decouples the correlation between samples and estimator after comparing first-order optimality conditions of the initial optimization with two leaving-one-out version; CGMT simplifies the first-order optimality conditions by making some relaxation of the PO problem; AMP solves the first order optimality conditions directly. All their asymptotic behaviours are characterized by the corresponding SEs respectively. The equivalence between these SEs sheds us light on looking for a more comprehensive theories to explain this intriguing phenomenon.

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

## A PROOF OF PROPOSITION 2.3

By the following linear parameter transformation:

$$\boldsymbol{w} := \boldsymbol{\beta} - \boldsymbol{\beta}^*,$$

the M-estimator optimization problem becomes:

$$\min_{\boldsymbol{w}} \frac{1}{n} \sum_{i=1}^{n} \rho(\epsilon_i - \mathbf{x}_i^T \boldsymbol{w}). \tag{22}$$

Introducing the Lagrange multiplier leads to:

$$\min_{\boldsymbol{w},\boldsymbol{v}} \max_{\boldsymbol{u}} \frac{1}{n} \sum_{i=1}^{n} \rho(v_i) + \frac{1}{\sqrt{n}} u_i (v_i - \epsilon_i + \mathbf{x}_i^T \boldsymbol{w})$$

where $\boldsymbol{u} := (u_1, ..., u_n); \boldsymbol{v} = (v_1, ..., v_n)$.

Then we rewrite it in the matrix form:

$$\min_{\boldsymbol{w},\boldsymbol{v}} \max_{\boldsymbol{u}} \frac{1}{\sqrt{n}} \boldsymbol{u}^T \mathbf{X} \boldsymbol{w} + \frac{1}{n} \sum_{i=1}^{n} \rho(v_i) + \frac{1}{\sqrt{n}} (\boldsymbol{u}^T \boldsymbol{v} - \boldsymbol{u}^T \boldsymbol{\epsilon})$$

where $\boldsymbol{\epsilon} := (\epsilon_1, ..., \epsilon_n)^T$. This is just the PO problem in CGMT.

Denote $\tilde{\mathbf{X}} = \sqrt{n} \mathbf{X}, \tilde{\boldsymbol{w}} = \frac{\boldsymbol{w}}{\sqrt{n}}$, then we have $\tilde{X}_{i,j} \overset{i.i.d.}{\sim} \mathcal{N}(0,1), \tilde{\boldsymbol{w}} = \frac{\boldsymbol{\beta} - \boldsymbol{\beta}^*}{\sqrt{n}}$. This means $\|\boldsymbol{w}\|^2$ is just the MSE of interest and $\tilde{\mathbf{X}}$ is a standard Gaussian matrix composed of iid standard normal variable.

However, in the following, we rewrite $\tilde{\mathbf{X}}, \tilde{\boldsymbol{w}}$ as $\mathbf{X}, \boldsymbol{w}$ respectively for the simplicity of notation.

Using CGMT about $\tilde{\mathbf{X}}$ as in (Salehi et al., 2019) by Corollary 3 in it, then the AO problem associated to it is the following min-max problem:

$$\min_{\boldsymbol{w},\boldsymbol{v}} \max_{\boldsymbol{u}} \frac{1}{\sqrt{n}} (\|\boldsymbol{u}\|_2 \boldsymbol{g}^T \boldsymbol{w} + \|\boldsymbol{w}\|_2 \boldsymbol{h}^T \boldsymbol{u}) + \frac{1}{n} \sum_{i=1}^{n} \rho(v_i) + \frac{1}{\sqrt{n}} (\boldsymbol{u}^T \boldsymbol{v} - \boldsymbol{u}^T \boldsymbol{\epsilon})$$

where $\boldsymbol{g} \in \mathbb{R}^d$ and $\boldsymbol{h} \in \mathbb{R}^n$ have i.i.d. $\mathcal{N}(0,1)$ entries.

Let $\|\boldsymbol{w}\|_2 = \tau_3$, note that now

$$\tau_3^2 = \frac{1}{n} \|\boldsymbol{\beta} - \boldsymbol{\beta}^*\|^2 \tag{23}$$

which is just the MSE.

Then the optimization becomes:

$$\min_{\tau_3,\boldsymbol{v}} \max_{\boldsymbol{u}} \frac{1}{\sqrt{n}} (-\tau_3 \|\boldsymbol{u}\|_2 \|\boldsymbol{g}\|_2 + \tau_3 \boldsymbol{h}^T \boldsymbol{u}) + \frac{1}{n} \sum_{i=1}^{n} \rho(v_i) + \frac{1}{\sqrt{n}} (\boldsymbol{u}^T \boldsymbol{v} - \boldsymbol{u}^T \boldsymbol{\epsilon}).$$

Letting $\|\boldsymbol{u}\|_2 = \mu$, then We have the following optimization:

$$\min_{\tau_3,\boldsymbol{v}} \max_{\mu>0} -\frac{\tau_3 \mu}{\sqrt{n}} \|\boldsymbol{g}\|_2 + \frac{1}{n} \sum_{i=1}^{n} \rho(v_i) + \frac{\mu}{\sqrt{n}} \|\tau \boldsymbol{h} + \boldsymbol{v} - \boldsymbol{\epsilon}\|_2. \tag{24}$$

equivalently:

$$\min_{\tau_3,\boldsymbol{v}} \max_{\mu>0} \frac{1}{n} \sum_{i=1}^{n} \rho(v_i) + \mu \left( \frac{1}{\sqrt{n}} \|\tau_3 \boldsymbol{h} + \boldsymbol{v} - \boldsymbol{\epsilon}\|_2 - \frac{\tau_3}{\sqrt{n}} \|\boldsymbol{g}\|_2 \right). \tag{25}$$

In order to make the $||\tau_3 \boldsymbol{h} + \boldsymbol{v} - \boldsymbol{\epsilon}||_2$ separable, we use the following optimization:

$$x = \min_{\alpha > 0} \frac{\alpha}{2} + \frac{x^2}{2\alpha}$$

for any $x$ and $\alpha > 0$. Replace $x$ by $\frac{1}{\sqrt{n}}||\tau_3 \boldsymbol{h} + \boldsymbol{v} - \boldsymbol{\epsilon}||_2$, the optimization problem (25) becomes:

$$\min_{\tau_3, \boldsymbol{v}} \max_{\mu} -\frac{\tau_3 \mu}{\sqrt{n}}||\boldsymbol{g}||_2 + \frac{1}{n}\sum_{i=1}^{n} \rho(v_i) + \mu(\max_{\alpha > 0} \frac{\alpha}{2} + \frac{1}{2\alpha n}||\tau_3 \boldsymbol{h} + \boldsymbol{v} - \boldsymbol{\epsilon}||_2^2).$$

Then what we want to do the scalarization procedure: make the optimization about $\boldsymbol{v}$ becoming optimization about a scalar. First, flipping in the order of min-max by (Thrampoulidis et al., 2018):

$$\min_{\tau_3, \alpha} \max_{\mu} \frac{\alpha\mu}{2} - \frac{\tau_3\mu}{\sqrt{n}}||\boldsymbol{g}||_2 + \frac{1}{n}[\min_{\boldsymbol{v}} \sum_{i=1}^{n} \rho(v_i) + \frac{\mu}{2\alpha}(\tau_3 h_i + v_i - \epsilon_i)^2]. \tag{26}$$

Introducing Moreau envelope function $M_\rho(x; t)$, the optimization problem (26) becomes:

$$\min_{\tau_3, \alpha} \max_{\mu} \frac{\alpha\mu}{2} - \frac{\tau_3\mu}{\sqrt{n}}||\boldsymbol{g}||_2 + \frac{1}{n}\sum_{i=1}^{n} M_\rho(\epsilon_i - \tau_3 h_i; \alpha/\mu).$$

By Lemma 9 in Appendix A in (Thrampoulidis et al., 2018), considering asymptotic $n, p \to \infty, p/n \to \kappa^*$ leads to:

$$\frac{\alpha\mu}{2} - \frac{\tau_3\mu}{\sqrt{n}}||\boldsymbol{g}||_2 + \frac{1}{n}\sum_{i=1}^{n} M_\rho(\epsilon_i - \tau_3 h_i; \alpha/\mu) \overset{a.s.}{\to} \frac{\alpha\mu}{2} - \tau_3\mu\sqrt{\kappa_*} + \mathbb{E}M_\rho(W_3 - \tau_3 Z_3; \alpha/\mu)$$

where $Z_3 \sim \mathcal{N}(0, 1)$ is independent of everything else.

Introduce $W_3 \sim P_\epsilon$. Then, asymptotically, we can deal with the following problem:

$$\min_{\tau_3, \alpha} \max_{\mu} \frac{\alpha\mu}{2} - \tau_3\mu\sqrt{\kappa_*} + \mathbb{E}M_\rho(W_3 + \tau_3 Z; \alpha/\mu). \tag{27}$$

Denoting the objective function of (27) by $\phi$, then since $\phi$ is convex about $(\tau_3, \alpha)$ and concave about $\mu$, the saddle point of $\phi$ can be precisely characterized by its first order optimality condition:

$$\frac{\partial\phi}{\partial\mu} = 0 \Rightarrow \frac{\alpha}{2} - \tau_3\sqrt{\kappa_*} - \frac{\alpha}{\mu^2}\mathbb{E}[\frac{\partial M_\rho}{\partial t}(W_3 + \tau_3 Z_3; \alpha/\mu)] = 0$$

$$\frac{\partial\phi}{\partial\tau} = 0 \Rightarrow -\mu\sqrt{\kappa_*} + \mathbb{E}[Z_3 \frac{\partial M_\rho}{\partial x}(W_3 + \tau_3 Z_3; \alpha/\mu)] = 0$$

$$\frac{\partial\phi}{\partial\alpha} = 0 \Rightarrow \frac{\mu}{2} + \frac{1}{\mu}\mathbb{E}[\frac{\partial M_\rho}{\partial t}(W_3 + \tau_3 Z_3; \alpha/\mu)] = 0.$$

Combining with (23) completes the proof.

## B    EQUIVALENCE OF SEs OF M-ESTIMATOR FROM AMP, LOO, CGMT

Recall that the SEs of M-estimator from LOO are:

$$1 - \kappa_* = \mathbb{E}[\frac{\partial Prox_\rho}{\partial x}(W_1 + \tau_1 Z_1; \lambda_1)]$$
$$\kappa_*\tau_1^2 := \mathbb{E}[W_1 + \tau_1 Z_1 - Prox_\rho(W_1 + \tau_1 Z_1; \lambda_1)]^2. \tag{28}$$

SEs of M-estimator from AMP are:

$$\tau_2^2 = \frac{1}{\kappa_*}\lambda_2^2\mathbb{E}[\frac{\partial M_\rho}{\partial x}(W_2 + \tau_2 Z_2; \lambda_2)]^2$$
$$\kappa_* = \lambda_2\mathbb{E}[\frac{\partial^2 M_\rho}{\partial x^2}(W_2 + \tau_2 Z_2; \lambda_2)]. \tag{29}$$

Next, we show that these two sets of SEs are equivalent.

First, Simple calculation leads to:

$$\frac{\partial M_\rho(x,t)}{\partial x} = \rho'(Prox_\rho(x;t))$$

$$\frac{\partial^2 M_\rho(x,t)}{\partial x^2} = \rho''(Prox_\rho(x;t))\frac{\partial Prox_\rho(x;t)}{\partial x} = \frac{\rho''(Prox_\rho(x;t))}{1 + t\rho''(Prox_\rho(x;t))} \qquad (30)$$

$$\frac{x - Prox_\rho(x;t)}{t} = \rho'(Prox_\rho(x;t)).$$

Combine this, SEs from AMP can be rewritten as:

$$\tau_2^2 = \frac{1}{\kappa_*} \lambda_2^2 \mathbb{E}[\rho'(Prox_\rho(W_2 + \tau_2 Z_2; \lambda_2))]^2$$

$$\kappa_* = \lambda_2 \mathbb{E}\Big[\frac{\rho''(Prox_\rho(W_2 + \tau_2 Z_2; \lambda_2))}{1 + \lambda_2 \rho''(Prox_\rho(W_2 + \tau_2 Z_2; \lambda_2))}\Big]. \qquad (31)$$

which verify that SEs from LOO and SEs from AMP are equivalent.

Next we prove equivalence of SEs from AMP and SEs from CGMT by parameter transformations suggested.

Recall the SEs from CGMT are:

$$\frac{\alpha}{2} - \tau_3\sqrt{\kappa_*} - \frac{\alpha}{\mu^2}\mathbb{E}[\frac{\partial M_\rho}{\partial t}(W_3 + \tau_3 Z_3; \alpha/\mu)] = 0$$

$$-\mu\sqrt{\kappa_*} + \mathbb{E}[Z_3\frac{\partial M_\rho}{\partial x}(W_3 + \tau_3 Z_3; \alpha/\mu)] = 0$$

$$\frac{\mu}{2} + \frac{1}{\mu}\mathbb{E}[\frac{\partial M_\rho}{\partial t}(W_3 + \tau_3 Z_3; \alpha/\mu)] = 0.$$

Let $b = \frac{\alpha}{\mu}$, then we have:

$$\frac{\mu b}{2} - \tau_3\sqrt{\kappa_*} - \frac{\alpha}{\mu^2}\mathbb{E}[\frac{\partial M_\rho}{\partial t}(W_3 + \tau_3 Z_3; b)] = 0$$

$$-\mu\sqrt{\kappa_*} + \mathbb{E}[Z_3\frac{\partial M_\rho}{\partial x}(W_3 + \tau_3 Z_3; b)] = 0$$

$$\frac{\mu}{2} + \frac{1}{\mu}\mathbb{E}[\frac{\partial M_\rho}{\partial y}(W + \tau Z; b)] = 0.$$

Comparing these results leads to:

$$\mathbb{E}[\frac{\partial M_\rho}{\partial x}(W_3 + \tau_3 Z_3; b)]^2 = \frac{\tau_3^2 \kappa_*}{b^2} \mathbb{E}[\frac{\partial^2 M_\rho}{\partial x^2}(W_3 + \tau_3 Z_3; b)] = \frac{\kappa_*}{b}. \qquad (32)$$

Combining stein lemma and

$$\frac{\partial M_\rho}{\partial t}(W_3 + \tau_3 Z_3; b) = -\frac{1}{2}[\frac{\partial M_\rho}{\partial x}(W_3 + \tau_3 Z_3; b)]^2. \qquad (33)$$

completes our proof.

## C  EQUIVALENCE OF SES OF LASSO FROM AMP AND CGMT

In this proof, we refer to the technique developed in (Donoho & Montanari, 2016). The Lasso problem solves

$$\arg\min_{\boldsymbol{\beta}} \frac{\lambda_*}{n}\|\boldsymbol{\beta}\|_1 + \frac{1}{2n}\|\mathbf{y} - \mathbf{X}\boldsymbol{\beta}\|^2. \tag{34}$$

Notice that $\|\mathbf{y} - \mathbf{X}\boldsymbol{\beta}\|^2 = \sum_{i=1}^n \left[(x_i^T\boldsymbol{\beta})^2 - 2y_i x_i^T\boldsymbol{\beta} + y_i^2\right]$ . The optimization can be transformed into

$$\arg\min_{\boldsymbol{\beta}} \frac{\lambda_*}{n}\|\boldsymbol{\beta}\|_1 + \frac{1}{n}\sum_{i=1}^n \left[\frac{1}{2}(x_i^T\boldsymbol{\beta})^2 - y_i x_i^T\boldsymbol{\beta}\right]. \tag{35}$$

In the following proof, we consider a more general optimization than Lasso:

$$\arg\min_{\boldsymbol{\beta}} \frac{\lambda_*}{n}f(\boldsymbol{\beta}) + \frac{1}{n}\sum_{i=1}^n \left[\mathbf{1}^T\boldsymbol{\rho}(\mathbf{u}) - y_i x_i^T\boldsymbol{\beta}\right] \tag{36}$$

where $\boldsymbol{\rho}(\cdot)$ and $f(\cdot)$ are 'separable' in the sense that there exist scalar functions $\rho(\cdot), \tilde{f}(\cdot)$ so that $\boldsymbol{\rho}(\cdot)$ and $f(\cdot)$ can be expressed as the following form: $\boldsymbol{\rho}(\mathbf{x}) = (\rho(x_1), \cdots, \rho(x_d))^T$ and $f(\mathbf{x}) = \sum_{i=1}^d \tilde{f}(x_i)$. In particular, in Lasso, $\rho(t) = \frac{1}{2}t^2$ and $f(\mathbf{x}) = \|\mathbf{x}\|_1$.

In order to apply CGMT, we introduce a new variable $\boldsymbol{u}$ and have following optimization

$$\min_{\boldsymbol{\beta},\mathbf{u}} \frac{\lambda_*}{n}f(\boldsymbol{\beta}) + \frac{1}{n}\left(\mathbf{1}^T\rho(\mathbf{u}) - y^T\mathbf{u}\right)$$
$$\text{s.t.} \quad \mathbf{u} = \mathbf{X}\boldsymbol{\beta} = \frac{1}{\sqrt{n}}\mathbf{H}^*\boldsymbol{\beta} \tag{37}$$

where $\mathbf{H}^* = \sqrt{n}\mathbf{X} \in \mathbb{R}^{n\times d}$ and hence $\mathbf{H}_{ij}^* \overset{\text{i.i.d.}}{\sim} \mathcal{N}(0,1)$. By using Lagrange multiplier we can rewrite (37) as a min-max optimization:

$$\min_{\boldsymbol{\beta}\in\mathbb{R}^d,\mathbf{u}\in\mathbb{R}^n} \max_{\boldsymbol{v}} \frac{1}{n}\mathbf{1}^T\rho(\mathbf{u}) - \frac{1}{n}y^T\mathbf{u} + \frac{\lambda_*}{n}\|\boldsymbol{\beta}\|_1 + \frac{1}{n}\boldsymbol{v}^T(\mathbf{u} - \frac{1}{\sqrt{n}}\mathbf{H}^*\boldsymbol{\beta}). \tag{38}$$

Denote $P = \frac{\boldsymbol{\beta}^*(\boldsymbol{\beta}^*)^T}{\|\boldsymbol{\beta}^*\|^2}$ as the projection matrix of true signal $\boldsymbol{\beta}^*$ and $P^\perp = I_d - P$ as the orthogonal complement. To apply CGMT, we need first decompose $\mathbf{H}^*$ into

$$\mathbf{H}_1^* = \mathbf{H}^* \cdot P, \quad \mathbf{H}_2^* = \mathbf{H}^* \cdot P^\perp$$
$$\mathbf{H}^* = \mathbf{H}_1^* + \mathbf{H}_2^*. \tag{39}$$

In addition, Recalling the linear model (3) we have $y = \mathbf{X}\boldsymbol{\beta}^* + \boldsymbol{\epsilon} = \frac{1}{\sqrt{n}}\mathbf{H}^*\boldsymbol{\beta}^* = \frac{1}{\sqrt{n}}\mathbf{H}_1^*\boldsymbol{\beta}^*$. Hence (38) can be rewritten as

$$\min_{\boldsymbol{\beta}\in\mathbb{R}^d,\mathbf{u}\in\mathbb{R}^n} \max_{\boldsymbol{v}} \frac{1}{n}\mathbf{1}^T\rho(\mathbf{u}) - \frac{1}{n}y^T\mathbf{u} + \frac{\lambda_*}{n}f(\boldsymbol{\beta}) + \frac{1}{n}\boldsymbol{v}^T(\mathbf{u} - \frac{1}{\sqrt{n}}\mathbf{H}_1^*\boldsymbol{\beta}) - \frac{1}{n\sqrt{n}}\boldsymbol{v}^T\mathbf{H}_2^*\boldsymbol{\beta}. \tag{40}$$

By using CGMT for $\boldsymbol{v}^T\mathbf{H}_2^*\boldsymbol{\beta}$ as in (Salehi et al., 2019) by Corollary 3 in it, the corresponding AO of (40) is

$$\min_{\boldsymbol{\beta}\in\mathbb{R}^d,\mathbf{u}\in\mathbb{R}^n} \max_{\boldsymbol{v}} \frac{1}{n}\mathbf{1}^T\rho(\mathbf{u}) - \frac{1}{n}y^T\mathbf{u} + \frac{\lambda_*}{n}f(\boldsymbol{\beta}) + \frac{1}{n}\boldsymbol{v}^T(\mathbf{u} - \frac{1}{\sqrt{n}}\mathbf{H}_1^*\boldsymbol{\beta})$$
$$- \frac{1}{n\sqrt{n}}\left(\boldsymbol{v}^T h\|P^\perp\boldsymbol{\beta}\|_2 + \|\boldsymbol{v}\|_2 g^T P^\perp\boldsymbol{\beta}\right) \tag{41}$$

where $h \sim \mathcal{N}(0, I_n)$ and $g \sim \mathcal{N}(0, I_d)$ are two independent gaussian vectors.

We first consider the maximization with respect to the direction of $v$. The part related to $v$ in optimization (41) is:

$$\max_{\boldsymbol{v}\in\mathbb{R}^n} \frac{1}{n\sqrt{n}}\|\boldsymbol{v}\|_2 g^T P^\perp \boldsymbol{\beta} + \frac{1}{n}\boldsymbol{v}^T\left(\mathbf{u} - \frac{1}{\sqrt{n}}\mathbf{H}_1^*\boldsymbol{\beta} - \frac{1}{\sqrt{n}}h\|P^\perp\boldsymbol{\beta}\|_2\right). \tag{42}$$

Denoting $r := \frac{\|\boldsymbol{v}\|_2}{\sqrt{n}}$ and maximizing along the direction of $v$ give

$$\max_{r\geq 0} r\left(\frac{1}{n}g^T P^\perp \boldsymbol{\beta} + \|\frac{1}{\sqrt{n}}\mathbf{u} - \frac{1}{n}\mathbf{H}_1^*\boldsymbol{\beta} - \frac{\|P^\perp\boldsymbol{\beta}\|_2}{n}h\|_2\right). \tag{43}$$

Inserting this into (41) gives

$$\min_{\boldsymbol{\beta},\mathbf{u}} \max_{r\geq 0} \frac{1}{n}\mathbf{1}^T\rho(\mathbf{u}) - \frac{1}{n}y^T\mathbf{u} + \frac{\lambda_*}{n}f(\boldsymbol{\beta}) + r\left(\frac{1}{n}g^T P^\perp\boldsymbol{\beta} + \left\|\frac{1}{\sqrt{n}}\mathbf{u} - \frac{1}{n}\mathbf{H}_1^*\boldsymbol{\beta} - \frac{\|P^\perp\boldsymbol{\beta}\|_2}{n}h\right\|_2\right). \tag{44}$$

In addition, introduce $\boldsymbol{\mu}$ to replace $\boldsymbol{\beta}$ in $\|\boldsymbol{\beta}\|_1$ Lagrangian and $\boldsymbol{w}$. Then (44) can be rewritten as,

$$\min_{\substack{\mathbf{u}\in\mathbb{R}^n;\\ \boldsymbol{\beta},\boldsymbol{\mu}\in\mathbb{R}^d}} \max_{\substack{r\geq 0;\\ \boldsymbol{w}\in\mathbb{R}^d}} \frac{1}{n}\mathbf{1}^T\rho(\mathbf{u}) - \frac{1}{n}y^T\mathbf{u} + \frac{\lambda_*}{n}f(\boldsymbol{\mu}) + r\left(\frac{1}{n}g^T P^\perp\boldsymbol{\beta}\right)$$
$$+ r\left\|\frac{1}{\sqrt{n}}\mathbf{u} - \frac{1}{n}\mathbf{H}_1^*\boldsymbol{\beta} - \frac{\|P^\perp\boldsymbol{\beta}\|_2}{n}h\right\|_2 + \frac{1}{d}\boldsymbol{w}^T(\boldsymbol{\mu} - \boldsymbol{\beta}). \tag{45}$$

We define $\alpha = \frac{\boldsymbol{\beta}^T\boldsymbol{\beta}^*}{\|\boldsymbol{\beta}^*\|_2^2}$, $\sigma = \frac{1}{\sqrt{n}}\|P^\perp\boldsymbol{\beta}\|_2$ and $\mathbf{q} = \frac{\mathbf{H}^*\boldsymbol{\beta}^*}{r_{1*}\sqrt{n}}$ where $r_{1*} = \frac{\|\boldsymbol{\beta}^*\|_2}{\sqrt{n}}$. Then $\mathbf{q}$ is a standard Gaussian vector and

$$\frac{1}{n}\mathbf{H}_1^*\boldsymbol{\beta} = \frac{1}{n}\mathbf{H}^*(P\boldsymbol{\beta}) = \frac{\mathbf{H}^*}{n}\cdot\alpha\boldsymbol{\beta}^* \overset{d}{=} \frac{\alpha}{n}r_{1*}\sqrt{n}\mathbf{q}. \tag{46}$$

Decomposing $\boldsymbol{w} = (P + P^\perp)\boldsymbol{w}$, then the last item in (45) can be rewritten as

$$\frac{1}{d}\boldsymbol{w}^T(\boldsymbol{\mu} - \boldsymbol{\beta}) = \frac{1}{d}(P\boldsymbol{w})^T\boldsymbol{\mu} + \frac{1}{d}(P^\perp\boldsymbol{w})^T\boldsymbol{\mu} - \frac{1}{d}(P\boldsymbol{w})^T\boldsymbol{\beta} - \frac{1}{d}(P^\perp\boldsymbol{w})^T\boldsymbol{\beta}. \tag{47}$$

Inserting (46) and (47) into (45), we have,

$$\min_{\substack{\mathbf{u}\in\mathbb{R}^n;\\ \boldsymbol{\beta},\boldsymbol{\mu}\in\mathbb{R}^d}} \max_{\substack{r\geq 0;\\ \boldsymbol{w}\in\mathbb{R}^d}} \frac{1}{n}\mathbf{1}^T\rho(\mathbf{u}) - \frac{1}{n}y^T\mathbf{u} + \frac{\lambda_*}{n}f(\boldsymbol{\mu}) + r\left(\frac{1}{n}g^T P^\perp\boldsymbol{\beta}\right) - \frac{1}{d}(P^\perp\boldsymbol{w})^T\boldsymbol{\beta}$$
$$+ r\left\|\frac{1}{\sqrt{n}}\mathbf{u} - \frac{\alpha r_{1*}}{\sqrt{n}}\mathbf{q} - \frac{\sigma}{\sqrt{n}}h\right\|_2 + \frac{1}{d}(P\boldsymbol{w})^T\boldsymbol{\mu} + \frac{1}{d}(P^\perp\boldsymbol{w})^T\boldsymbol{\mu} - \frac{1}{d}(P\boldsymbol{w})^T\boldsymbol{\beta}. \tag{48}$$

Then we can fix $P\boldsymbol{\beta}$ and consider the minimization along the direction of $P^\perp\boldsymbol{\beta}$. Considering the optimization related to $P^\perp\boldsymbol{\beta}$, we have

$$\min_{P^\perp\boldsymbol{\beta}} \frac{r}{n}g^T P^\perp\boldsymbol{\beta} - \frac{1}{d}(P^\perp\boldsymbol{w})^T\boldsymbol{\beta} = \min_{P^\perp\boldsymbol{\beta}}(\frac{r}{n}g^T - \frac{1}{d}\boldsymbol{w}^T)P^\perp\boldsymbol{\beta}$$
$$= -\|P^\perp\boldsymbol{\beta}\|_2 \cdot \|\frac{r}{n}P^\perp g - \frac{1}{d}P^\perp\boldsymbol{w}\|_2 \tag{49}$$
$$= -\sigma \cdot \|\frac{r}{\sqrt{n}}P^\perp g - \sqrt{\frac{1}{d\kappa_*}}P^\perp\boldsymbol{w}\|_2.$$

Notice that (48) reaches optimal when $\boldsymbol{\mu} = \boldsymbol{\beta}$. Inserting (49) into (48) leads to

$$
\min_{\substack{\mathbf{u} \in \mathbb{R}^n, \boldsymbol{\mu} \in \mathbb{R}^d; \\ \alpha \in \mathbb{R}, \sigma \geq 0}} \max_{\substack{r \geq 0; \\ \boldsymbol{w} \in \mathbb{R}^d}} \frac{1}{n} \mathbf{1}^T \rho(\mathbf{u}) - \frac{1}{n} y^T \mathbf{u} + \frac{\lambda_*}{n} f(\boldsymbol{\mu}) - \sigma \cdot \| \frac{r}{\sqrt{n}} P^\perp g - \sqrt{\frac{1}{d\kappa_*}} P^\perp \boldsymbol{w} \|
$$
$$
+ r \left\| \frac{1}{\sqrt{n}} \mathbf{u} - \frac{\alpha r_{1*}}{\sqrt{n}} \mathbf{q} - \frac{\sigma}{\sqrt{n}} h \right\| + \frac{1}{d} (P^\perp \boldsymbol{w})^T \boldsymbol{\mu}.
\tag{50}
$$

For the simplifying procedures in the following steps in our analysis, we change $\| \cdot \|_2 \to \| \cdot \|_2^2$ by

$$
rx = \min_{v \geq 0} \frac{r}{2v} + \frac{rv}{2} x^2
$$
$$
-\sigma x = \max_{\tau \geq 0} -\frac{\sigma}{2\tau} - \frac{\sigma\tau}{2} x^2.
\tag{51}
$$

Applying (51), we are able to rewrite (50) as

$$
\min_{\mathbf{u} \in \mathbb{R}^n, \boldsymbol{\mu} \in \mathbb{R}^d; \alpha \in \mathbb{R}, v, \sigma \geq 0} \max_{\substack{\boldsymbol{w} \in \mathbb{R}^d; \\ r, \tau \geq 0}} \frac{1}{n} \mathbf{1}^T \rho(\mathbf{u}) - \frac{1}{n} y^T \mathbf{u} + \frac{\lambda_*}{n} f(\boldsymbol{\mu}) - \frac{\sigma}{2\tau} - \frac{\sigma\tau}{2} \| \frac{r}{\sqrt{n}} P^\perp g - \sqrt{\frac{1}{d\kappa_*}} P^\perp \boldsymbol{w} \|^2
$$
$$
+ \frac{r}{2v} + \frac{rv}{2} \left\| \frac{1}{\sqrt{n}} \mathbf{u} - \frac{\alpha r_{1*}}{\sqrt{n}} \mathbf{q} - \frac{\sigma}{\sqrt{n}} h \right\|^2 + \frac{1}{d} (P^\perp \boldsymbol{w})^T \boldsymbol{\mu}.
\tag{52}
$$

**Optimization with respect to $\boldsymbol{w}$**: Next we consider the maximization with respect to $\boldsymbol{w}$. We first extract the item related to $\boldsymbol{w}$ in (52) and apply the completion of squares:

$$
\max_{\boldsymbol{w}} - \frac{\sigma\tau}{2} \left\| \frac{r}{\sqrt{n}} P^\perp g - \sqrt{\frac{1}{d\kappa_*}} P^\perp \boldsymbol{w} \right\|^2 + \frac{1}{d} (P^\perp \boldsymbol{w})^T \boldsymbol{\mu}
$$
$$
= \max_{\boldsymbol{w}} - \frac{\sigma\tau}{2} \left\| \frac{r}{\sqrt{n}} P^\perp g - \sqrt{\frac{1}{d\kappa_*}} P^\perp \boldsymbol{w} + \frac{1}{\sqrt{d/\kappa_*}\sigma\tau} P^\perp \boldsymbol{\mu} \right\|^2
\tag{53}
$$
$$
+ \frac{1}{2n\sigma\tau} \left\| P^\perp \boldsymbol{\mu} + \sigma\tau r P^\perp g \right\|^2 - \frac{\sigma\tau r^2}{2n} \| P^\perp g \|^2.
$$

1) For the last item in (53), since $g \sim \mathcal{N}(0, I_d)$ and $P^\perp$ is a $(n-1)$-dimensional projection matrix, we derive that $\| P^\perp g \|_2^2 \sim \| d(0, (P^\perp)^2) \|_2^2 \overset{d}{=} \chi_{d-1}^2$ and

$$
\frac{\sigma\tau r^2}{2n} \| P^\perp g \|^2 \overset{a.s.}{\to} \frac{\sigma\tau r^2 \kappa_*}{2}.
\tag{54}
$$

2) Since $P^\perp = I_d - P$, the second item in (53) can be rewritten as

$$
\frac{1}{n} \left\| P^\perp \boldsymbol{\mu} + \sigma\tau r P^\perp g \right\|^2 = \frac{1}{n} \| \boldsymbol{\mu} + \sigma\tau r g \|^2 - \frac{1}{n} \| P\boldsymbol{\mu} \|^2
$$
$$
- \frac{(\sigma\tau r)^2}{n} \| Pg \|^2 - \frac{2\sigma\tau r}{n} (Pg)^T \boldsymbol{\mu}.
\tag{55}
$$

The last two items of (55) can be omitted in the limit of $d, n \to \infty$ because $\frac{\| Pg \|^2}{n} = O_p(\frac{1}{n})$ and $\frac{1}{n}(Pg)^T \boldsymbol{\mu} = O_p(\frac{1}{\sqrt{n}})$. The second item of (55) is $\frac{1}{n} \| P\boldsymbol{\mu} \|^2 = \frac{1}{n} \| P\boldsymbol{\beta} \|^2 = \alpha^2 r_{1*}^2$ by definition.

3) The first item in (53) reaches 0 when maximizing $\boldsymbol{w}$.

The optimization (52) now can be rewritten as

$$\min_{\substack{\mathbf{u}\in\mathbb{R}^n, \boldsymbol{\mu}\in\mathbb{R}^d; \\ \alpha\in\mathbb{R}, v, \sigma\geq 0}} \max_{r,\tau\geq 0} \frac{1}{n}\mathbf{1}^T\rho(\mathbf{u}) - \frac{1}{n}y^T\mathbf{u} + \frac{\lambda_*}{n}f(\boldsymbol{\mu}) - \frac{\sigma}{2\tau} + \frac{1}{2n\sigma\tau}\|\boldsymbol{\mu} + \sigma\tau rg\|^2 - \frac{\alpha^2 r_{1*}^2}{2\sigma\tau} - \frac{\sigma\tau r^2\kappa_*}{2}$$
$$+ \frac{r}{2v} + \frac{rv}{2}\left\|\frac{1}{\sqrt{n}}\mathbf{u} - \frac{\alpha r_{1*}}{\sqrt{n}}\mathbf{q} - \frac{\sigma}{\sqrt{n}}h\right\|^2. \tag{56}$$

**Optimization respect to $\boldsymbol{\mu}$:** Consider the items related to $\boldsymbol{\mu}$ in (56)

$$\min_{\boldsymbol{\mu}\in\mathbb{R}^d} \frac{\lambda_*}{n}f(\boldsymbol{\mu}) + \frac{1}{2n\sigma\tau}\|\boldsymbol{\mu} + \sigma\tau rg\|_2^2 \tag{57}$$
$$s.t. \qquad \boldsymbol{\mu} = \boldsymbol{\beta}.$$

Notice that $g\sim\mathcal{N}(0, I_d)$, $\|\boldsymbol{\mu} + \sigma\tau rg\|_2^2 \stackrel{d}{=} \|\boldsymbol{\mu} - \sigma\tau rg\|_2^2$. We rewrite (57) as

$$\min_{\boldsymbol{\mu}\in\mathbb{R}^d} \frac{\lambda_*}{n}f(\boldsymbol{\mu}) + \frac{1}{2n\sigma\tau}\|\boldsymbol{\mu} - \sigma\tau rg\|_2^2 \tag{58}$$
$$s.t. \qquad \frac{1}{n}\boldsymbol{\beta}^{*T}\boldsymbol{\mu} = \frac{1}{n}\boldsymbol{\beta}^{*T}\boldsymbol{\beta} = \frac{1}{n}n\alpha r_{1*}^2 = \alpha r_{1*}^2.$$

Introducing Lagrangian $\theta$, (58) can be rewrite as

$$\min_{\boldsymbol{\mu}\in\mathbb{R}^d} \max_{\theta\in\mathbb{R}} \frac{1}{2n\sigma\tau}\|\boldsymbol{\mu} - \sigma\tau rg\|^2 + \frac{\lambda_*}{n}f(\boldsymbol{\mu}) - \frac{\theta}{n}\boldsymbol{\beta}^{*T}\boldsymbol{\mu} + \alpha\theta r_{1*}^2. \tag{59}$$

Applying the completion of squares to 1,3 items in (59) we have,

$$\frac{1}{2n\sigma\tau}\|\boldsymbol{\mu} - \sigma\tau rg\|^2 - \frac{\theta}{n}\boldsymbol{\beta}^{*T}\boldsymbol{\mu} = \frac{1}{2n\sigma\tau}\|\boldsymbol{\mu} - \sigma\tau rg - \theta\sigma\tau\boldsymbol{\beta}^*\|^2$$
$$- \frac{(\theta\sigma\tau)^2}{2n\sigma\tau}\|\boldsymbol{\beta}^*\|^2 - \frac{\theta r\sigma^2\tau^2}{2n\sigma\tau}g^T\boldsymbol{\beta}^*. \tag{60}$$

The third item can be omitted since $\frac{g^T\boldsymbol{\beta}^*}{n} = O_p(\frac{1}{\sqrt{n}})$ and the second item has limit $-\frac{(\theta\sigma\tau)^2}{2n\sigma\tau}\|\boldsymbol{\beta}^*\|^2 \to -\frac{(\theta\sigma\tau)^2}{2n\sigma\tau}\cdot nr_{1*}^2 = -\frac{\sigma\tau\theta^2 r_{1*}^2}{2}$. Hence we rewrite right side of (60) as

$$\frac{1}{2n\sigma\tau}\|\boldsymbol{\mu} - \sigma\tau rg\|^2 - \frac{\theta}{n}\boldsymbol{\beta}^{*T}\boldsymbol{\mu} = \frac{1}{2n\sigma\tau}\|\boldsymbol{\mu} - \sigma\tau rg - \theta\sigma\tau\boldsymbol{\beta}^*\|^2 - \frac{\sigma\tau\theta^2 r_{1*}^2}{2}. \tag{61}$$

Next, denote $\tilde{f}(x)$ as single-entry form of $f(x)$. We can rewrite the (61) in terms of Moreau envelope entry-wisely as follows

$$\min_{\boldsymbol{\mu}\in\mathbb{R}^d} \max_{\theta\in\mathbb{R}} \frac{1}{2n\sigma\tau}\|\boldsymbol{\mu} - \sigma\tau rg\|^2 + \frac{\lambda_*}{n}f(\boldsymbol{\mu}) - \frac{\theta}{n}\boldsymbol{\beta}^{*T}\boldsymbol{\mu} + \alpha\theta r_{1*}^2$$
$$= \max_{\theta} \frac{1}{n}M_{\lambda_*\tilde{f}}(\sigma\tau(rg + \theta\boldsymbol{\beta}^*); \sigma\tau) + \alpha\theta r_{1*}^2 - \frac{\sigma\tau\theta^2 r_{1*}^2}{2}. \tag{62}$$

Substituting (62) in (56) we have,

$$\min_{\substack{\mathbf{u}\in\mathbb{R}^n; \\ \alpha\in\mathbb{R}, \sigma, v\geq 0}} \max_{\substack{r,\tau\geq 0; \\ \theta\in\mathbb{R}}} \frac{1}{n}\mathbf{1}^T\rho(\mathbf{u}) - \frac{1}{n}y^T\mathbf{u} + \frac{rv}{2}\left\|\frac{1}{\sqrt{n}}\mathbf{u} - \frac{\alpha r_{1*}}{\sqrt{n}}\mathbf{q} - \frac{\sigma}{\sqrt{n}}h\right\|^2 - \frac{\sigma}{2\tau} - \frac{\alpha^2 r_{1*}^2}{2\sigma\tau}$$
$$- \frac{\sigma\tau r^2\kappa_*}{2} + \frac{r}{2v} + \frac{1}{n}M_{\lambda_*f}(\sigma\tau(rg + \theta\boldsymbol{\beta}^*); \sigma\tau) + \alpha\theta r_{1*}^2 - \frac{\sigma\tau\theta^2 r_{1*}^2}{2}. \tag{63}$$

**Optimization respect to $u$**: First we consider the items related to $u$. The optimization is

$$\min_{\mathbf{u}\in\mathbb{R}^n}\frac{1}{n}\mathbf{1}^T\rho(\mathbf{u})-\frac{1}{n}y^T\mathbf{u}+\frac{rv}{2}\left\|\frac{1}{\sqrt{n}}\mathbf{u}-\frac{\alpha r_{1*}}{\sqrt{n}}\mathbf{q}-\frac{\sigma}{\sqrt{n}}h\right\|^2. \tag{64}$$

Applying the completion of squares we have,

$$-\frac{1}{n}y^T\mathbf{u}+\frac{rv}{2}\left\|\frac{1}{\sqrt{n}}\mathbf{u}-\frac{\alpha r_{1*}}{\sqrt{n}}\mathbf{q}-\frac{\sigma}{\sqrt{n}}h\right\|^2=\frac{rv}{2}\left\|\frac{1}{\sqrt{n}}\mathbf{u}-\frac{\alpha r_{1*}}{\sqrt{n}}\mathbf{q}-\frac{\sigma}{\sqrt{n}}h-\frac{1}{rv\sqrt{n}}y\right\|^2 \\ -\frac{1}{2rvn}\|y\|^2-\frac{r_{1*}\alpha}{n}y^T\mathbf{q}-\frac{\sigma}{n}y^Th. \tag{65}$$

Using the strong law of large numbers we have,

$$-\frac{1}{2rvn}\|y\|^2\overset{a.s.}{\to}-\frac{r_{1*}^2+\sigma_*^2}{2rv} \\ -\frac{r_{1*}\alpha}{n}y^T\mathbf{q}\overset{a.s.}{\to}-r_{1*}^2\alpha \\ -\frac{\sigma}{n}y^Th\overset{a.s.}{\to}0. \tag{66}$$

Next, by substituting (65), (66) in (64), we can rewritten the optimization as,

$$\min_{\mathbf{u}\in\mathbb{R}^n}\frac{1}{n}\mathbf{1}^T\rho(\mathbf{u})-\frac{1}{n}y^T\mathbf{u}+\frac{rv}{2}\left\|\frac{1}{\sqrt{n}}\mathbf{u}-\frac{\alpha r_{1*}}{\sqrt{n}}\mathbf{q}-\frac{\sigma}{\sqrt{n}}h\right\|^2 \\ =\min_{\mathbf{u}\in\mathbb{R}^n}\frac{1}{n}\mathbf{1}^T\rho(\mathbf{u})+\frac{rv}{2}\left\|\frac{1}{\sqrt{n}}\mathbf{u}-\frac{\alpha r_{1*}}{\sqrt{n}}\mathbf{q}-\frac{\sigma}{\sqrt{n}}h-\frac{1}{rv\sqrt{n}}y\right\|^2 \\ -\frac{r_{1*}^2+\sigma_*^2}{2rv}-r_{1*}^2\alpha. \tag{67}$$

Then we can rewrite (67) in terms of Moreau envelope,

$$\min_{\mathbf{u}\in\mathbb{R}^n}\frac{1}{n}\mathbf{1}^T\rho(\mathbf{u})+\frac{rv}{2}\left\|\frac{1}{\sqrt{n}}\mathbf{u}-\frac{\alpha r_{1*}}{\sqrt{n}}\mathbf{q}-\frac{\sigma}{\sqrt{n}}h-\frac{1}{rv\sqrt{n}}y\right\|^2 \\ =\frac{1}{n}M_{\rho(\cdot)}(\alpha r_{1*}\mathbf{q}+\sigma h+\frac{1}{rv}y;\frac{1}{rv}). \tag{68}$$

Substituting (67), (68) in (63) we have

$$\min_{\substack{\alpha\in\mathbb{R};\\\sigma,v\geq0}}\max_{\substack{r,\tau\geq0;\\\theta\in\mathbb{R}}}\quad\frac{1}{n}M_\rho(\alpha r_{1*}\mathbf{q}+\sigma h+\frac{1}{rv}y;\frac{1}{rv})-\frac{r_{1*}^2+\sigma_*^2}{2rv}-r_{1*}^2\alpha \\ -\frac{\sigma}{2\tau}-\frac{\alpha^2 r_{1*}^2}{2\sigma\tau}-\frac{\sigma\tau r^2\kappa_*}{2}+\frac{r}{2v}+\alpha\theta r_{1*}^2-\frac{\sigma\tau\theta^2 r_{1*}^2}{2} \\ +\frac{1}{n}M_{\lambda_* f}(\sigma\tau(rg+\theta\boldsymbol{\beta}^*);\sigma\tau). \tag{69}$$

**Final scalarization**: Using the strong law of large number ($\mathbf{q},h,y,g,\boldsymbol{\beta}^*$ are entry-wise i.i.d.), we can rewrite (69) as

$$\min_{\substack{\alpha \in \mathbb{R}, \\ \sigma, v \geq 0}} \max_{\substack{r, \tau \geq 0; \\ \theta \in \mathbb{R}}} \quad -\frac{\sigma}{2\tau} - \frac{\alpha^2 r_{1*}^2}{2\sigma\tau} - \frac{\sigma\tau r^2 \kappa_*}{2} + \frac{r}{2v} + \alpha\theta r_{1*}^2 - \frac{\sigma\tau\theta^2 r_{1*}^2}{2} - \frac{r_{1*}^2 + \sigma_*^2}{2rv} - r_{1*}^2 \alpha$$

$$+ \mathbb{E}\left[ M_\rho\left(\alpha r_{1*} Z_1 + \sigma Z_2 + \frac{1}{rv}(r_{1*} Z_1 + \sigma_* Z_3); \frac{1}{rv}\right)\right] \tag{70}$$

$$+ \mathbb{E}\left[M_{\lambda_* \tilde{f}}(\sigma\tau(rZ + \theta b_0); \sigma\tau)\right] \cdot \frac{d}{n}$$

where $Z_1, Z_2, Z \sim \mathcal{N}(0, 1)$, $\sigma_* Z_3 \sim P_\epsilon$ and $b_0 \sim \Pi$ are all independent.

For LASSO, $\tilde{f}(x) = |x|$. The Moreau envelope $M_{\lambda\tilde{f}}(\cdot; \cdot)$ has property:

$$M_{\lambda_* \tilde{f}}(\sigma\tau(rZ + \theta b_0); \sigma\tau) = \sigma\tau \cdot M_{\lambda_* \tilde{f}}(rZ + \theta b_0; 1). \tag{71}$$

Besides, for $\rho(x) = \frac{1}{2}x^2$ in LASSO, the Moreau envelope $M_\rho(\cdot; \cdot)$ has explict form:

$$M_\rho(v; t) = \frac{v^2}{2(t+1)} \tag{72}$$

and the second last item of (70) can be simplified to:

$$\mathbb{E}\left[M_{\rho(\cdot)}\left(\alpha r_{1*} Z_1 + \sigma Z_2 + \frac{1}{rv}(r_{1*} Z_1 + \sigma_* Z_3); \frac{1}{rv}\right)\right] = \mathbb{E}\left[\frac{\left((\alpha + \frac{1}{rv})r_{1*} Z_1 + \sigma Z_2 + \frac{\sigma_*}{rv} Z_3\right)^2}{2(\frac{1}{rv} + 1)}\right]$$

$$= \frac{r_{1*}^2(\alpha rv + 1)^2 + r^2 v^2 \sigma^2 + \sigma_*^2}{2(1 + rv)rv}. \tag{73}$$

In order to simplify (73), we denote $\lambda = \frac{1}{rv}$ in place of $v$. At this time, $\min_{v \geq 0}$ is replaced by $\max_{\lambda \geq 0}$ and

$$\frac{r_{1*}^2(\alpha rv + 1)^2 + r^2 v^2 \sigma^2 + \sigma_*^2}{2(1 + rv)rv} = \frac{(\alpha + \lambda)^2 r_{1*}^2 + \sigma^2 + \sigma_*^2 \lambda^2}{2(\lambda + 1)}. \tag{74}$$

Substituting (74) in (70) we have the final optimization for LASSO:

$$\min_{\substack{\alpha \in \mathbb{R}; \\ \sigma \geq 0}} \max_{\substack{r, \tau, \lambda \geq 0; \\ \theta \in \mathbb{R}}} \quad -\frac{\sigma}{2\tau} - \frac{\alpha^2 r_{1*}^2}{2\sigma\tau} - \frac{\sigma\tau r^2 \kappa_*}{2} + \frac{r^2 \lambda}{2} + \alpha\theta r_{1*}^2 - \frac{\sigma\tau\theta^2 r_{1*}^2}{2} - \frac{(r_{1*}^2 + \sigma_*^2)\lambda}{2} - r_{1*}^2 \alpha$$

$$+ \frac{(\alpha + \lambda)^2 r_{1*}^2 + \sigma^2 + \sigma_*^2 \lambda^2}{2(\lambda + 1)} + \mathbb{E}\left[M_{\lambda_* \tilde{f}}(rZ + \theta b_0; 1)\right] \cdot \sigma\tau\kappa_* \tag{75}$$

which is a smooth function with respect to $\alpha, \sigma, r, \tau, \lambda, \theta$. Let $\phi$ denote the objective function of (75).

**Deriving SEs from function** $\phi$: The SEs are given by the first order optimality conditions of $\phi$:

1) For $\frac{\partial\phi}{\partial\alpha} = 0$:

$$-\frac{\alpha}{\sigma\tau} + \theta - 1 + \frac{\alpha + \lambda}{\lambda + 1} = 0. \tag{76}$$

2) For $\frac{\partial\phi}{\partial\sigma} = 0$:

$$-\frac{1}{2\tau} - \frac{\tau r^2 \kappa_*}{2} + \frac{r_{1*}^2 \alpha^2}{2\sigma^2\tau} - \frac{r_{1*}^2 \tau\theta^2}{2} + \tau\kappa_* E[M_{\lambda_* \tilde{f}}(rZ + \theta b_0; 1)] + \frac{\sigma}{\lambda + 1} = 0. \tag{77}$$

3) For $\frac{\partial \phi}{\partial \lambda} = 0$:

$$r^2 - r_{1*}^2 - \sigma_*^2 + \frac{2[(\alpha + \lambda)r_{1*}^2 + \lambda \sigma_*^2]}{\lambda + 1} - \frac{(\alpha + \lambda)^2 r_{1*}^2 + \sigma^2 + \lambda^2 \sigma_*^2}{(\lambda + 1)^2} = 0. \qquad (78)$$

4) For $\frac{\partial \phi}{\partial \theta} = 0$:

$$r_{1*}^2 \alpha - \sigma \tau \kappa_* \mathbb{E}[b_0(Prox_{\lambda_* \tilde{f}}(rZ + \theta b_0))] = 0. \qquad (79)$$

where we use the definition $\mathbb{E}[b_0^2] = E_\Pi X^2 = r_{1*}^2$

5) For $\frac{\partial \phi}{\partial r} = 0$:

$$-\sigma \tau r \kappa_* + r\lambda + \sigma \tau \kappa_* \mathbb{E}[(rZ + \theta b_0 - Prox_{\lambda_* \tilde{f}}(rZ + \theta b_0; 1))Z] = 0. \qquad (80)$$

Since $\mathbb{E}[Z^2] = 1, \mathbb{E}[Zb_0] = 0$. For any function $\tilde{f}(x)$, the Moreau envelope and proximal operator of $\tilde{f}$ stratifies

$$\frac{\partial}{\partial x} M_{\tilde{f}}(x; t) = \frac{x - Prox_{\tilde{f}}(x; t)}{t}. \qquad (81)$$

Then we rewrite the equation (80) as

$$r\lambda - \sigma \tau \kappa_* \mathbb{E}[Prox_{\lambda_* \tilde{f}}(rZ + \theta b_0; 1)Z] = 0. \qquad (82)$$

Using Stein lemma,

$$\begin{aligned} \mathbb{E}[Prox_{\lambda_* \tilde{f}}(rZ + \theta b_0; 1)Z] &= \mathbb{E}[r \frac{\partial Prox_{\lambda_* \tilde{f}}(rZ + \theta b_0; 1)}{\partial x}] \\ &= \mathbb{E}[\frac{\partial Prox_{\lambda_* \tilde{f}}(rZ + \theta b_0; 1)}{\partial Z}], \end{aligned} \qquad (83)$$

(82) can be rewritten as

$$\lambda = \sigma \tau \kappa_* \mathbb{E}[\frac{\partial Prox_{\lambda_* \tilde{f}}(rZ + \theta b_0; 1))}{\partial x}]. \qquad (84)$$

6) For $\frac{\partial \phi}{\partial \tau} = 0$:

$$\frac{\sigma}{2\tau^2} - \frac{\sigma r^2 \kappa_*}{2} + \frac{r_{1*}^2 \alpha^2}{2\sigma \tau^2} - \frac{r_{1*}^2 \sigma \theta^2}{2} + \sigma \kappa_* \mathbb{E}[M_{\lambda_* \tilde{f}}(rZ + \theta b_0; 1)] = 0. \qquad (85)$$

For any function $\tilde{f}(x)$, the Moreau envelope and proximal operator of $\tilde{f}$ stratifies

$$M_{\lambda_* \tilde{f}}(x; b) = \lambda_* M_{\tilde{f}}(x; \lambda_* b) = \frac{x^2}{2b} - \frac{[Prox_{\tilde{f}}(x; \lambda_* b)]^2}{2b}, \quad \forall \lambda_*, b > 0, x \in \mathbb{R} \qquad (86)$$

Using this property, we can rewrite (85) as

$$\frac{\sigma}{2\tau^2} - \frac{\sigma r^2 \kappa_*}{2} + \frac{r_{1*}^2 \alpha^2}{2\sigma \tau^2} - \frac{r_{1*}^2 \sigma \theta^2}{2} + \sigma \kappa_* \mathbb{E}[\frac{(rZ + \theta b_0)^2}{2} - \frac{[Prox_{\lambda_* \tilde{f}}(rZ + \theta b_0; 1)]^2}{2}] = 0 \qquad (87)$$

i.e.,

$$\frac{\sigma}{2\tau^2} + \frac{r_{1*}^2 \alpha^2}{2\sigma \tau^2} - \frac{\sigma \kappa_*}{2} E[(Prox_{\lambda_* \tilde{f}}(rZ + \theta b_0; 1))^2] = 0. \qquad (88)$$

Similarly, the equation (77) derived by $\frac{\partial \phi}{\partial \sigma} = 0$ can be rewritten as

$$-\frac{1}{2\tau} - \frac{\tau r^2 \kappa_*}{2} + \frac{r_{1*}^2 \alpha^2}{2\sigma^2 \tau} - \frac{r_{1*}^2 \tau \theta^2}{2} + \tau \kappa_* E\Big[\frac{(rZ + \theta b_0)^2}{2} - \frac{[Prox_{\lambda_* \tilde{f}}(rZ + \theta b_0; 1)]^2}{2}\Big] + \frac{\sigma}{\lambda + 1} = 0$$

(89)

i.e.,

$$-\frac{1}{2\tau} + \frac{r_{1*}^2 \alpha^2}{2\sigma^2 \tau} - \frac{\tau \kappa_*}{2} E[(Prox_{\lambda_* \tilde{f}}(rZ + \theta b_0; 1))^2] + \frac{\sigma}{\lambda + 1} = 0.$$

(90)

Hence we get the SEs by summarizing equations (76), (90), (78), (79), (84), (88)

$$
\begin{aligned}
0 &= -\frac{\alpha}{\sigma\tau} + \theta - 1 + \frac{\alpha + \lambda}{\lambda + 1} \\
0 &= -\frac{1}{2\tau} + \frac{r_{1*}^2 \alpha^2}{2\sigma^2 \tau} - \frac{\tau \kappa_*}{2} E[(Prox_{\lambda_* \tilde{f}}(rZ + \theta b_0; 1))^2] + \frac{\sigma}{\lambda + 1} \\
0 &= r^2 - r_{1*}^2 - \sigma_*^2 + \frac{2[(\alpha + \lambda)r_{1*}^2 + \lambda\sigma_*^2]}{\lambda + 1} - \frac{(\alpha + \lambda)^2 r_{1*}^2 + \sigma^2 + \lambda^2 \sigma_*^2}{(\lambda + 1)^2} \\
0 &= r_{1*}^2 \alpha - \sigma\tau\kappa_* E[b_0(Prox_{\lambda_* \tilde{f}}(rZ + \theta b_0; 1))] \\
\lambda &= \sigma\tau\kappa_* E\Big[\frac{\partial Prox_{\lambda_* \tilde{f}}(rZ + \theta b_0; 1))}{\partial x}\Big] \\
0 &= \frac{\sigma}{2\tau^2} + \frac{r_{1*}^2 \alpha^2}{2\sigma\tau^2} - \frac{\sigma\kappa_*}{2} E[(Prox_{\lambda_* \tilde{f}}(rZ + \theta b_0; 1))^2].
\end{aligned}
$$

(91)

regarding $(\alpha, \sigma, \lambda, \theta, r, \tau)$.

Since $r_*^2 = \mathbb{E}_{b_0 \sim \Pi} b_0^2 = \frac{r_{1*}^2}{\kappa_*}$, the SEs (91) can be rewritten as

$$0 = -\frac{\alpha}{\sigma\tau} + \theta - 1 + \frac{\alpha + \lambda}{\lambda + 1} \tag{92a}$$

$$0 = -\frac{1}{2\tau} + \frac{r_*^2 \kappa_* \alpha^2}{2\sigma^2 \tau} - \frac{\tau \kappa_*}{2} E[(Prox_{\lambda_* \tilde{f}}(rZ + \theta b_0; 1))^2] + \frac{\sigma}{\lambda + 1} \tag{92b}$$

$$0 = r^2 - r_*^2 \kappa_* - \sigma_*^2 + \frac{2[(\alpha + \lambda)r_*^2 \kappa_* + \lambda\sigma_*^2]}{\lambda + 1} - \frac{(\alpha + \lambda)^2 r_*^2 \kappa_* + \sigma^2 + \lambda^2 \sigma_*^2}{(\lambda + 1)^2} \tag{92c}$$

$$0 = r_*^2 \kappa_* \alpha - \sigma\tau\kappa_* E[b_0(Prox_{\lambda_* \tilde{f}}(rZ + \theta b_0; 1))] \tag{92d}$$

$$\lambda = \sigma\tau\kappa_* E\Big[\frac{\partial Prox_{\lambda_* \tilde{f}}(rZ + \theta b_0; 1))}{\partial x}\Big] \tag{92e}$$

$$0 = \frac{\sigma}{2\tau^2} + \frac{r_*^2 \kappa_* \alpha^2}{2\sigma\tau^2} - \frac{\sigma\kappa_*}{2} E[(Prox_{\lambda_* \tilde{f}}(rZ + \theta b_0; 1))^2]. \tag{92f}$$

regarding $(\alpha, \sigma, \lambda, \theta, r, \tau)$. This is equivalent to the SEs (12) except for the notations are slightly different.

## C.1 EQUIVALENCE OF SES

We first rewrite $r$, $\tau$, $Z$ and $b_0$ in (92) to $\gamma_2$, $\tau_2$ and $Z_2$ $\beta_2$ respectively, the equation (92e) becomes

$$\lambda = \sigma\tau_2\kappa_* \mathbb{E}\Big[\frac{\partial Prox_{\tilde{f}}(\gamma_2 Z_2 + \theta\beta_2; \lambda_*)}{\partial x}\Big] = \sigma\tau_2\kappa_* \mathbb{E}[\eta'(\gamma_2 Z_2 + \theta\beta_2; \lambda_*)] \tag{93}$$

for $\tilde{f}(x) = |x|$.

Then we simplify the SEs (92). Consider equations (92b) and (92f) and we have

$$\sigma\tau_2 = \lambda + 1, \tag{94}$$

substituting (94) in (92a) we have

$$\theta = \frac{1}{\lambda + 1} = \frac{1}{\sigma \tau_2}. \tag{95}$$

For the second equation of AMP in (11), which is

$$\gamma_1 = \kappa_*(\gamma_1 + \lambda_*)\mathbb{E}[\eta'(\beta_1 + \tau_1 Z_1; \lambda_* + \gamma_1)], \tag{96}$$

it is obviously equivalent to the equation (92e) if we have parameter transformations $\tau_1 = \frac{\gamma_2}{\theta}$ and $\gamma_1 = \lambda_*(\frac{1}{\theta} - 1)$. A property of $\eta(\cdot; \cdot)$ is used for the equivalence:

$$\eta'(cx; ct) = \eta'(x; t), \quad \forall c > 0. \tag{97}$$

Using the parameter transformations mentioned above and denote $W = \eta(\beta_1 + \tau_1 Z_1; \lambda_* + \gamma_1)$, the first equation of AMP in (11) can be rewritten as

$$\frac{\gamma_2^2}{\theta^2} = \sigma_*^2 + \kappa_* \mathbb{E}[W - \beta_2]^2 \tag{98}$$

i.e.,

$$\frac{\gamma_2^2}{\theta^2} = \sigma_*^2 + \kappa_*(\mathbb{E}[W^2] + \mathbb{E}[\beta_2^2] - 2\mathbb{E}[W\beta_2]). \tag{99}$$

Substituting (92d), (92f) and $\mathbb{E}(\beta_2^2) = r_*^2$, we have

$$\begin{aligned}
\mathbb{E}(W^2) &= \mathbb{E}(\eta^2(\beta_1 + \tau_1 Z_1; \lambda_* + \gamma_1)) \\
&= \mathbb{E}(\eta^2(\beta_1 + \frac{\gamma_2}{\theta} Z_1; \frac{\lambda_*}{\theta})) \\
&= \frac{1}{\theta^2} \mathbb{E}(\eta^2(\theta\beta_2 + \gamma_2 Z_2; \lambda_*)) \quad \text{(because } \eta(cx; ct) = c\eta(x; t)) \\
&= \frac{1}{\theta^2}\left[\frac{1}{\tau_2^2 \kappa_*} + \frac{r_*^2 \alpha^2}{\sigma^2 \tau_2^2}\right] \quad \text{(using (92f))}
\end{aligned} \tag{100}$$

$$\begin{aligned}
-2\mathbb{E}(W\beta_2) &= -2\mathbb{E}(\eta(\beta_1 + \tau_1 Z_1; \lambda_* + \gamma_1) \cdot \beta_1) \\
&= -\frac{2}{\theta}\mathbb{E}(\eta(\theta\beta_2 + \gamma_2 Z_2; \lambda_*) \cdot \beta_2) \\
&= -\frac{2}{\theta}\frac{r_*^2 \alpha}{\sigma \tau_2} \quad \text{(using (92d))},
\end{aligned}$$

then (99) can be rewritten as

$$\frac{r^2}{\theta^2} = \sigma_*^2 + \kappa_*\left(\frac{1}{\theta^2 \tau_2^2 \kappa_*} + \frac{r_*^2 \alpha^2}{\theta^2 \sigma^2 \tau_2^2} - \frac{2r_*^2 \alpha}{\theta \sigma \tau_2 + r_*^2}.\right) \tag{101}$$

Using (95) in (101) we have

$$r^2(\sigma\tau_2)^2 = \sigma_*^2 + \sigma^2 + \kappa_*(\alpha - 1)^2 r_*^2. \tag{102}$$

Besides, for CGMT, the equation (92c) can be written as

$$(\lambda + 1)^2 r^2 - (\alpha - 1)^2 r_*^2 \kappa_* - \sigma^2 - \sigma_*^2 = 0. \tag{103}$$

Using (95) in (103) we have

$$(\sigma\tau_2)^2 r^2 = \sigma_*^2 + (\alpha - 1)^2 r_*^2 \kappa_* + \sigma^2. \tag{104}$$

The equation (102) and (104) are equivalent. Hence the equations (92d), (92f) and (92c) of CGMT can be shown to be a decomposition of the first equation of AMP in (11) after some parameter transformations. In conclusion we prove the equivalence between SEs from CGMT and AMP in Lasso framework.

# D RELAXATION PHENOMENON

## D.1 RELAXATION PHENOMENON OF M-ESTIMATOR

Notice that (24) is equivalent to

$$\min_{\tau_3, \boldsymbol{v}} \frac{1}{n} \sum_{i=1}^{n} \rho(v_i) \quad s.t. \frac{1}{\sqrt{n}} ||\tau_3 \boldsymbol{h} + \boldsymbol{v} - \boldsymbol{\epsilon}||_2 \leq \frac{\tau_3}{\sqrt{n}} ||\boldsymbol{g}||_2. \tag{105}$$

On the other hand, by rotation invariance of Gaussian distribution, (22) becomes

$$\min_{||\boldsymbol{w}||_2, \boldsymbol{v}} \frac{1}{n} \sum_{i=1}^{n} \rho(v_i) \quad s.t. \, ||\boldsymbol{v} - \boldsymbol{\epsilon} + ||\boldsymbol{w}||_2 Z'|| = 0 \tag{106}$$

where $Z' \sim \mathcal{N}(0,1), \boldsymbol{\epsilon} := (\epsilon_1, ..., \epsilon_n)^T$. Comparing (105) with (106) can verify the relaxation phenomenon, i.e. the only difference between AO and PO is that the feasible region of PO is a subset of the feasible region of AO.

## D.2 RELAXATION PHENOMENON OF SUPPORT VECTOR MACHINE AND LOGISTIC REGRESSION

The relaxation phenomenon of support vector machine and logistic regression can be similarly shown as we have done in Appendix D.1, so we omit the proof here.

# E EQUIVALENCE OF SEs OF LOGISTIC REGRESSION FROM LOO AND CGMT

By doing the following parameter transformations:

$$\alpha_2 = \sqrt{\kappa_*} \alpha_1, \mu = r_* \sigma, \lambda_2 = \lambda_1,$$

SEs of CGMT become:

$$\begin{aligned} 0 &= \mathbb{E}[Vl'(Prox_{\lambda_1 l}(\sqrt{\kappa_*}\alpha_1 Z + r_*\sigma V)))] \\ \kappa_*^2(\alpha_1)^2 &= (\lambda_1)^2 \mathbb{E}[(l'(Prox_{\lambda_1 l}(\sqrt{\kappa_*}\alpha_1 Z + r_*\sigma V)))^2] \\ \kappa_* &= \lambda_1 \mathbb{E}[\frac{l''(Prox_{\lambda_1 l}(\sqrt{\kappa_*}\alpha_1 Z + r_*\sigma V))}{1 + \lambda_1 l''(Prox_{\lambda_1 l}(\sqrt{\kappa_*}\alpha_1 Z + r_*\sigma V))}]. \end{aligned} \tag{107}$$

What we want to prove is that:

$$\begin{aligned} \mathbb{E}[(l'(prox_{\lambda_1 l}(\sqrt{\kappa_*}\alpha_1 Z + r_*\sigma V)))^2] &= \mathbb{E}[2\rho'(Q_1)(\rho'(prox_{\lambda_1 \rho}(Q_2)))^2] \\ 1 - \lambda_1 \mathbb{E}[\frac{l''(prox_{\lambda_1 l}(\sqrt{\kappa_*}\alpha_1 Z + r_*\sigma V))}{1 + \lambda_1 l''(prox_{\lambda_1 l}(\sqrt{\kappa_*}\alpha_1 Z + r_*\sigma V))}] &= \mathbb{E}[\frac{2\rho'(Q_1)}{1 + \lambda_1 \rho''(prox_{\lambda_1 \rho}(Q_2))}] \\ E[Vl'(prox_{\lambda_1 l}(\sqrt{\kappa_*}\alpha_1 Z + r_*\sigma V))] &= c\mathbb{E}[\rho'(Q_1)Q_1\rho'(prox_{\lambda_1 \rho}(Q_2))] \end{aligned}$$

where $c$ is a constant.

First, we verify the following identity:

$$\mathbb{E}[(l'(prox_{\lambda_1 l}(\sqrt{\kappa_*}\alpha_1 Z + r_*\sigma V)))^2] = \mathbb{E}[2\rho'(Q_1)(\rho'(prox_{\lambda_1 \rho}(Q_2)))^2]. \tag{108}$$

Note that

$$\begin{aligned} l(t) &= \rho(-t) \\ l'(t) &= -\rho'(-t) \\ l''(t) &= \rho(t) \\ Prox_{\lambda_1 l}(z) &= -Prox_{\lambda_1 \rho}(-z) \end{aligned} \tag{109}$$

and the probability density function (pdf) of $V = GY$ is

$$P_V(v) = \frac{1}{\sqrt{2\pi}} e^{-\frac{v^2}{2}} \frac{2}{1 + e^{-r_* v}},$$

we have:

$$\mathbb{E}[(l'(Prox_{\lambda_1 l}(\sqrt{\kappa_*}\alpha_1 Z + r_* \sigma V)))^2] = \iint (l'(Prox_{\lambda_1 l}(\sqrt{\kappa_*}\alpha_1 h + r_* \sigma v)))^2 P_Z(h) P_V(v) dh dv$$

$$= \iint 2(\rho'(Prox_{\lambda_1 \rho}(-\sqrt{\kappa_*}\alpha_1 h - r_* \sigma v)))^2 \frac{1}{2\pi} e^{-\frac{h^2 + v^2}{2}} \rho'(r_* v) dh dv \tag{110}$$

where $P_Z(h) := \frac{1}{\sqrt{2\pi}} e^{-\frac{h^2}{2}}$ is the pdf of $Z$. Meanwhile,

$$\mathbb{E}[2\rho'(Q_1)(\rho'(prox_{\lambda_1 \rho}(Q_2)))^2] = \iint 2\rho'(q_1)(\rho'(prox_{\lambda_1 \rho}(q_2)))^2 P_{Q_1, Q_2}(q_1, q_2) dq_1 dq_2. \tag{111}$$

Now we introduce the following parameter transformations: $q_1 = r_* v, q_2 = -\sqrt{\kappa_*}\alpha_1 h - r_* \sigma v$.
Then (111) becomes

$$\iint \sqrt{\kappa_*}\alpha_1 r_* * 2\rho'(r_* v)(\rho'(prox_{\lambda_1 \rho}(-\sqrt{\kappa_*}\alpha_1 h - r_* \sigma v)))^2 P_{Q_1, Q_2}(r_* v, -\sqrt{\kappa_*}\alpha_1 h - r_* \sigma v) dh dv.$$

In order to verify (108), we only need to prove that:

$$\frac{1}{2\pi} e^{-\frac{h^2 + v^2}{2}} = \sqrt{\kappa_*}\alpha_1 r_* P_{Q_1, Q_2}(r_* v, -\sqrt{\kappa_*}\alpha_1 h - r_* \sigma v).$$

Construct $Q_1', Q_2'$ as follows: assume $Z', V' \overset{i.i.d.}{\sim} \mathcal{N}(0,1)$ and

$$\begin{pmatrix} Q_1' \\ Q_2' \end{pmatrix} = \begin{pmatrix} 0 & r_* \\ -\sqrt{\kappa_*}\alpha_1 & -r_* \sigma \end{pmatrix} \begin{pmatrix} Z' \\ V' \end{pmatrix}.$$

We can easily verify that:

$$\mathbb{E}[(Q_1', Q_2')^T] = (0, 0)^T$$

$$Cov[(Q_1', Q_2')^T] = \begin{pmatrix} r_*^2 & -r_* \sigma r_* \\ -r_* \sigma r_* & \sqrt{\kappa_*}\alpha_1^2 + r_* \sigma^2 \end{pmatrix}$$

which means $(Q_1', Q_2')$ has identical distribution of $(Q_1, Q_2)$.
On the other hand, since

$$P_{Q_1', Q_2'}(q_1', q_2') dq_1' q_2' = P_{Z', V'}(h', v') dh' dv'$$

$$\frac{dq_1' q_2'}{dh' dv'} = r_* \sqrt{\kappa_*}\alpha_1,$$

$$q_1' = r_* v',$$

$$q_2' = -\sqrt{\kappa_*}\alpha_1 h' - r_* \sigma v',$$

we have:

$$\frac{1}{2\pi} e^{-\frac{h'^2 + v'^2}{2}} = \sqrt{\kappa_*}\alpha_1 r_* P_{Q_1', Q_2'}(r_* v', -\sqrt{\kappa_*}\alpha_1 h' - r_* \sigma v')$$

which completes our proof of (108).

Secondly, we prove:

$$1 - \lambda_1 \mathbb{E}\left[\frac{l''(prox_{\lambda_1 l}(\sqrt{\kappa_*}\alpha_1 Z + r_* \sigma V))}{1 + \lambda_1 l''(prox_{\lambda_1 l}(\sqrt{\kappa_*}\alpha_1 Z + r_* \sigma V))}\right] = \mathbb{E}\left[\frac{2\rho'(Q_1)}{1 + \lambda_1 \rho''(prox_{\lambda_1 \rho}(Q_2))}\right]. \tag{112}$$

Left hand side (LHS) of (112) is

$$\mathbb{E}[\frac{1}{1+\lambda_1 l''(prox_{\lambda_1 l}(\sqrt{\kappa_*}\alpha_1 Z + r_*\sigma V))}] = \iint \frac{1}{1+\lambda_1 l''(prox_{\lambda_1 l}(\sqrt{\kappa_*}\alpha_1 h + r_*\sigma v))}P_Z(h)P_V(v)dhdv$$

$$= \iint \frac{1}{1+\lambda_1 l''(prox_{\lambda_1 l}(\sqrt{\kappa_*}\alpha_1 h + r_*\sigma v))}\frac{1}{2\pi}e^{-\frac{h^2+v^2}{2}}2\rho'(r_*v)dhdv$$

$$= \iint \frac{1}{1+\lambda_1 \rho''(prox_{\lambda_1 \rho}(-\sqrt{\kappa_*}\alpha_1 h - r_*\sigma v))}\frac{1}{2\pi}e^{-\frac{h^2+v^2}{2}}2\rho'(r_*v)dhdv.$$

(113)

Through parameter transformations $q_1 = r_*v, q_2 = -\sqrt{\kappa_*}\alpha_1 h - r_*\sigma v$, RHS of (112) becomes

$$\iint \frac{2\rho'(q_1)}{1+\lambda_2 \rho''(prox_{\lambda_1 \rho}(q_2))}P_{Q_1,Q_2}(q_1, q_2)dq_1 dq_2$$

$$= \iint \frac{2\rho'(r_*v)}{1+\lambda_1 \rho''(prox_{\lambda_1 \rho}(-\sqrt{\kappa_*}\alpha_1 h - r_*\sigma v))}P_{Q_1,Q_2}(r_*v, -\sqrt{\kappa_*}\alpha_1 h - r_*\sigma v)r_*\sqrt{\kappa_*}\alpha_1 dhdv.$$

Combining with

$$\frac{1}{2\pi}e^{-\frac{h^2+v^2}{2}} = P_{Q_1,Q_2}(r_*v, -\sqrt{\kappa_*}\alpha_1 h - r_*\sigma v)r_*\sqrt{\kappa_*}\alpha_1$$

completes the proof of (112).

The proof of

$$\mathbb{E}[Vl'(prox_{\lambda_1 l}(\sqrt{\kappa_*}\alpha_1 Z + r_*\sigma V))] = c\mathbb{E}[\rho'(Q_1)Q_1\rho'(prox_{\lambda_1 \rho}(Q_2))]$$

can be derived similarly. So we omit the proof here.

