# OpenReview forum: "Equivalence of State Equations from Different Methods in High-dimensional Regression"
_ICLR.cc/2022/Conference — ICLR 2022 Submitted_

### Official Review · Reviewer_e98M · 2021-10-30

**Correctness:** 4
**Technical Novelty And Significance:** 4
**Empirical Novelty And Significance:** Not applicable
**Recommendation:** 8
**Confidence:** 4

**Main Review:**

While the main results of the paper were believed to be true by many researchers, to my knowledge this is the first paper that lays out clearly what the equivalence is between the outcomes of these 3 techniques. Hence the contribution of the paper is undeniable.

However, on closer inspection, the paper does not provide any insight into why this equivalence holds, most of the explanations are cursory at best.

To my knowledge, the derivation from the CGMT technique is based solely on the fixed points of an optimization problem. This provides a somewhat geometric method since KKT conditions are essentially geometric in nature.
On the other hand, the AMP derivation is essentially carefully constructing a dynamical system that has an equivalent scalar dynamical system. That this dynamical system reaches the fixed point of an optimization problem is an added benefit. Thus AMP-based derivations are in principle based on some iterative decorrelation given by the Onsager correction. This ends up being some sort of adaptive curvature adjustment in an energy landscape.

There is little discussion about the LOO technique or insights related to it. The bit on page 8 is more confusing than clear.


**Summary Of The Paper:**

This paper compares the so-called "State Evolution" equations from 3 different derivations for the MSE of
high dimensional M-estimators in the proportional asymptotic regime (number of samples and features go to infinity at a fixed rate).
There are currently 3 well-known frameworks for analysis in this regime:
1. Approximate Message Passing (AMP)
2. Convex Gaussian Min-Max Theorem (CGMT)
3. Leave One Out (LOO)
The paper shows that there lies an equivalence between the results derived by the 3 methods via a parameter transformation.
Two important special cases of LASSO and Binary Logistic Regression are considered and the results are clearly stated.

**Summary Of The Review:**

The claims made in the paper are correct to the best of my knowledge.

The paper deserves to be published even though it lacks in several other ways, mainly because it is the first of its kind in establishing an equivalence between 3 different derivations for the asymptotic analysis of estimators.

This body of work can serve as a way for researchers using one method to check their predictions with another method using a parameter transformation suggested in this paper.

However, there is serious room for improvement in the presentation of the material discussed. The lack of insight provided is also a concern but perhaps can be clarified in future works.

---

> ### Author Response · Authors · 2021-11-21
> **Response to Reviewer e98M**
>
> Thanks for your summary and detailed comments! Your summary shows that
> you are a responsible reviewer and konw this field very well. We believe
> that your comments about our paper are significantly professional. The
> encouraging review indicates that you are $\textbf{deeply aware of the value of
> this paper}$. This review gives us great confidence and inspire us further
> research in this field.
>
> I would like to clarify your concerns in the following.
>
> Reviewer: On closer inspection, the paper does not provide any
> insight into why this equivalence holds, most of the explanations are
> cursory at best.
>
> Response: Thanks for the insightful questions. Indeed, this paper do not give too
> much insight into the reason for equivalence. The main purpose of this
> proposal is to communicate with other scholars and share our $\textbf{interesting
> obeservations}$. The insight of equivalence deserve further study and we
> will try our best to clarify it in the following work.
>
> However, we want to point out even the equivalence of SE is $\textbf{meaningful
> and significant}$:
>
> 1-The equivalence of SEs is the most $\textbf{direct, accessible, basic}$  way to
> establish the equivalence. And it is the $\textbf{basis}$ of more insightful
> equivalence.
>
> 2-Since parameters in different methods express different statistical
> meanings in their own ways. The parameter transformations can help us
> $\textbf{related the statistical meanings}$ in different methods, which can help us
> establish the equivalence of three methods. The strong similarity of
> parameter transformations in different settings(say, different theorems)
> can inspire us to explore equivalence of three methods dedicatedly,
> which is our further goal.
>
> Reviewer: There is little discussion about the LOO technique or
> insights related to it. The bit on page 8 is more confusing than
> clear.
>
> Response: Thank you for your comments and questions. We are so sorry the contents
> about LOO makes you confused. The confusion perhaps mainly because we
> have not given detailed explanation about this method and have omitted
> to many notations. In order to aviod similar confusion of readers, we
> have tried to provided more detailed disscussions ablout LOO.
>
> The new version of $\textbf{Deriving SEs from LOO}$ are as follows:
>
>
>  1- First, for the original optimization problem, LOO considers
>   first-order conditions of three cases: (1) keeping all observations
>   and predictors, corresponding solution is denoted as $\hat\beta$,
>   (2) leaving one predictor, corresponding solution is denoted as
>   $\hat\beta_{(-j)}$ and (3) leaving one predictor and one
>   observation, corresponding solution is denoted as
>   $\hat\beta_{(-i),(-j)}$
>
>  2- Two properties are derived from comparing three version of first-order
>   conditions: (1) The $i$-th fitted value $X_i\hat\beta$ has an
>   asymptotic expression composed of two independent random vectors
>   $X_{i,(-j)}$ and $\hat\beta_{(-i),(-j)}$. (2) Each coordinate
>   $\hat\beta_j$ can be written as a sum of $n$ random variables
>   which are asymptotically independent.
>
>   3-Using above two properties , $\hat\beta_j$ has the same distribution
>   as a combination of several scalar variables when $n,p\to \infty$.
>   Hence every statistic of $\hat\beta$ (such as expectation, variance
>   and first order condition of optimization) can be expressed by these
>   scalar variables, from which the SEs of $\hat\beta$ are derived.
>
>
> Reviewer:The paper $\textbf{deserves to be published}$ even though it lacks in
> several other ways, mainly because it is the first of its kind in
> establishing an equivalence between 3 different derivations for the
> asymptotic analysis of estimators.
>
> Response: Thank you for your recommandations and consideration. We feel very
> relieved you can realize the value of this paper and the difficulty of
> our research process. We will try our best to provide lacking contents in the following work.

---

### Official Review · Reviewer_12zP · 2021-11-01

**Correctness:** 3
**Technical Novelty And Significance:** 2
**Empirical Novelty And Significance:** Not applicable
**Recommendation:** 3
**Confidence:** 4

**Main Review:**

STRENGHTS

AMP, LOO and CGMT have been widely studied in the recent literature, and it has been already pointed out that, in some cases, results concerning the performance of a certain high-dimensional inference task can be obtained using different methods (with very different proof techniques). Thus, understanding the deep connections underlying this fact is certainly a worthwhile goal.

------------

WEAKNESSES

This paper falls short from this goal and does not offer much insight into *why* the fixed points are the same. The proofs of the equivalence of the fixed point are rather simple and they do not shed much light into the deep connections between e.g. AMP and CGMT. Proposition 3.2 appears to be novel, but its proof can be derived from existing work.

From a stylistic viewpoint, the paper contains several typos (especially in the appendix; e.g. Appendix D.2 is repeated twice). Thus, I would recommend the authors to perform a thorough check during the revision.

**Summary Of The Paper:**

This paper considers the state evolution equations obtained via different methods (AMP, LOO and CGMT), and it shows that their fixed points are the same for some high-dimensional inference problems (M-estimation, LASSO, and logistic regression).

Most of these state evolutions were obtained in recent work (which is properly cited here). An exception is Proposition 3.2, which gives the state evolution for CGMT in the case of LASSO. However, the proof of this result closely follows the existing literature. Consequently, the main novelty of this contribution is in showing that the fixed points (when unique) are the same for different methods.

**Summary Of The Review:**

While the paper focuses on a very interesting subject, unfortunately it does not seem to offer much insight into why the different state evolutions lead to the same fixed point. As a result, the contribution of this work appears to be clearly below the acceptance bar.

---

> ### Author Response · Authors · 2021-11-21
> **Response to Reviewer 12zP**
>
> Thanks for your precise summary and detailed comments. I would like to
> clarify your concerns in the following:
>
> Reviewer: This paper falls short from this goal and
> does not offer much insight into $\textbf{why}$ the fixed points are the
> same. This paper falls short from this goal and does not offer much insight into why the fixed points are the same.
>
> Response: Equivalence of SEs is only the first step and can be viewed as a basis
> of other deeper equivalence. The main purpose of this proposal is to
> communicate with other scholars and share our interesting obeservations.
> However, deeper equivalence, such as equivalence of different methods,
> is hard to clarify since the absense of related work. As has been
> figured out by reviewer e98M. We will clarify the deeper equivalence in
> the following work.
>
> Meanwhile, we want to figure out enen the equivalence of SE is
> $\textbf{meaningful and significant}$:
>
> 1-The equivalence of state equation is the most $\textbf{direct, accessible,
> basic}$ way to establish the equivalence. And it is the $\textbf{basis}$ of more
> insightful equivalence.
>
> 2-Since parameters in different methods express different statistical
> meanings in their own ways. The parameter transformations can help us
> $\textbf{related the statistical meanings}$ in different methods, which can help us
> establish the equivalence of three methods. The strong similarity of
> parameter transformations in different settings(say, different theorems)
> can inspire us to explore equivalence of three methods dedicatedly,
> which is our further goal.
>
> 3-To the equivalence of SEs itself, it can serve as a way for
> researchers using one method to $\textbf{check their predictions}$ with another
> method using a parameter transformation suggested in this paper, which
> has indicated in the comment of reviewer e98M.
>
> Reviewer: From a stylistic viewpoint, the paper contains several
> typos.Thus, I would recommend the authors to perform a thorough check during
> the revision.
>
> Response:Thank you for your suggestion. We have revised these mistakes and
> performed $\textbf{ a thorough check}$ during the revision.
>
> Reviewer: The proofs of the equivalence of the fixed point are
> rather simple:
>
> Response: Though the proof is not too complex, but it is $\textbf{not tirvial}$ obviously:
> proof of the equivalence needs a lot of calculations and a prior belief
> in the equivalence. Meanwhile, choice of parameter transformations is
> not too simple.

---

> > ### Comment · Reviewer_12zP · 2021-11-29
> > **Acknowledging the authors' feedback**
> >
> > I would like to thank the authors for their thoughtful reply. However, while the aim of the paper is certainly of interest, regretfully the novelty and significance of the results is clearly below bar, in my opinion. Some of these results were known in the literature and the mere proof of the equivalence of the SE equations is not enough to accept the paper. Furthermore, the paper should undergo a major stylistic revision before it is ready for publication. The feedback of the authors does not really address my criticism and does not convince me about the relevance of this contribution.

---

### Official Review · Reviewer_rXB2 · 2021-11-02

**Correctness:** 2
**Technical Novelty And Significance:** 1
**Empirical Novelty And Significance:** Not applicable
**Recommendation:** 1
**Confidence:** 5

**Main Review:**

The only strength for me is the a-priori good intention of the paper; as a specialist of the field I was a-priori interested by the paper's goal, but unfortunately it did not convince me at all.

The weaknesses are:

0) it is not written (nor very interesting) for the ICLR audience; it is a paper for specialists, on a a-priori interesting field to ICLR readers, but the way it is written makes it completely inaccessible to non-specialists.

1) It is badly written and full of typos. The authors should make a important effort to make it readable.

2) It clearly is oriented towards specialists of the field but most of the text is only recalling known results by others. "Pedagogical parts" like "Derived SEs from AMP", "Derived SEs from CGMT", "Derived SEs from LOO" are completely useless: a specialist does not need it (and anyway it is badly written I believe), and a non-specialist will get nothing out of it.

3) The main results are very restricted: a) they only considered M-estimators for linear regression (and to a lesser extend logistic regression) with random design; b) the results, being (sometimes quite straighforward) consequences of change of variables which link the non-linear equations derived from the 3 different perspectives, bring absolutely no insight to the reader, other than a formal equivalence.

4) The authors seem to miss important knowledge of the field: not a single paper from the community of physics of spin glasses is cited. But the leave-one-out approach is actually the cavity method from spin glasses with just another name. Having some knowledge on the cavity method, it is completely obvious that the SE equation derived from leave-one-out = cavity method yields he same as AMP, because AMP is just an approximation to Belief Propagation, which is derived from the cavity method. I mean: AMP or BP are just the algorithmic counterpart of the cavity method, and the cavity method is precisely designed to study the statistical properties of BP/AMP, and there are thus of course equivalent up ti simple transformations that simply depend of the details on how the computation have been performed. So to me, this part of the "equivalence results" was trivial and know for a very long time. Not everyone is familiar with these concepts and literature, so a-priori it may be interesting to show a relation between AMP and LOO, but it is not possible to do so by completely skipping all that is known..
Again, what is really not re-assuring, is that the authors did not even mention these know facts and related literature.
The part on the equivalence between CGMT and the rest seems more interesting to me. But in the appendix the authors repeat in their own way computations mostly known already in the literature (that they cite: Thrampoulidis et al, etc). And because their computations bring no additional insights, I don't see the added value (as such) of the contribution.

4) Whole chunk of related literature are completely skipped and not discussed or even cited; it misses too much here to provide a list

**Summary Of The Paper:**

This paper provides derivations of equivalences between what they refer to as "state evolution" (SE) equations derived from different perspectives: convex Gaussian min-max (CGMT), approximate message-passing (AMP) and leave-one-out (LOO) approaches. They focus on M-estimators for high-dimensional linear regression. The SE equations describe the mean-square error reconstruction of the M-estimator.

**Summary Of The Review:**

To summarize, I believe that despite the potential A-PRIORI interest by some specialised readers, the paper will not convince these specialist readers either. And it will noot bring much to non-specialists. It is badly written, and no real insights are provided. Many computations are just adaptations of things present elsewhere. Many known facts and equivalences are not cited nor even alluded to (again the equivalence between the AMP state evolution and LOO = cavity method is known to any physicist with some experience in spin glasses). The relevant literature is hugely missing.

I suggest the authors to completely re-write the paper, and consider instead a direct journal submission, I really do not see the point too submit this work to ICLR. And, importantly, to cite much more carefully and expmain more thoroughly what is new or not in the paper (as such, I do not believe that much is actually new).

---

> ### Author Response · Authors · 2021-11-21
> **Response to Reviewer rXB2**
>
> Thanks for your summary and detailed comments.  This shows you are a responsible reviewer and have great achievement in physics, especially, cavity method. We also believe you are a specialist in this field. I would like to clarify your concerns in the following:
>
> weakness 0:
>
> High dimensional statistical problem might be one of the most important topics in the learning theory. It provides us the prototype and tools to explore more complex models and data. Any developments of the theory and method in high dimensional (linear) regression and other statistical problems will enlarge our toolbox and improve our understandings.
>
> All the methods ( AMP, CGMT and LOO ) are designed for analyzing the high dimensional statistical problems. An intrinsic equivalence between these methods will help us understand the high dimensional statistical problem.
> We are sorry for that we are failure to provide such an intrinsic explanation and only report the equivalence `canonically' over the state equations. We do believe its development should be $\textbf{of the general interests }$ for the statistical learning community, such as Reviewer U2rr, Reviewer 12zP, Reviewer e98M.
>
> weakness 1:
>
> Thank you for your thorough reading! We are so sorry for the typos in
> last version of submitted paper and inconvenience they caused in your
> reading. We have $\textbf{carefully revised it thoroughly}$.
>
> weakness 2
>
> Indeed, there are some known results in this paper, which maybe make
> this paper looks lack of innovation at a first glance. However, this
> paper also has $\textbf{a lot of unknown results}$, such as theorems in this paper
> and SEs of lasso. Meanwhile, the aim of this paper is to establish the
> equivalence of three konwns methods. Recalling results of this field
> aims to explaining the equivalence (of SEs or methods) clearly. The
> Pedagogical parts are designed for the same reason and the
> self-containment.
>
> weakness 3
>
> a): The reason why this paper only focus on these optimization problems
> are as follows: The main purpose of this article is to $\textbf{propose an interesting
> phenomenon}$, not a complex technique. So we only focus our attention on
> sereval illustrative examples.  Our focus point can also promote $\textbf{accessiability.}$
>
>
> b): Though the proof is not too complex, it is not tirvial obviously:
> proof of the equivalence needs a lot of calculations and a prior belief
> in the equivalence. Meanwhile, choice of parameter transformations is
> not trival.
>
> Deeper equivalence, such as equivalence of different methods, is
> hard to clarify since the absense of related work, as have clarified by
> reviewer e98M. Equivalence of SEs maybe not too very difficult. But it
> is meaningful and significant:
>
> 1-The equivalence of state equation is the most $\textbf{direct, accessible,
> basic}$ way to establish the equivalence. And it is the $\textbf{basis}$ of more
> insightful equivalence.
>
> 2-Since parameters in different methods express different statistical
> meanings in their own ways. The parameter transformations can help us
> $\textbf{related the statistical meanings}$ in different methods, which can help us
> establish the equivalence of three methods. The strong similarity of
> parameter transformations in different settings(say, different theorems)
> can inspire us to explore equivalence of three methods dedicatedly,
> which is our further goal.
>
> 3-To the equivalence of SEs itself, it can serve as a way for
> researchers using one method to $\textbf{check their predictions}$ with another
> method using a parameter transformation suggested in this paper, which
> has indicated in the comment of reviewer e98M.
>
> weakness 4
>
> We are glad to get the response from experts like you. We are sorry for that we somehow missed a spate of literature from the statistical physics community. This is mainly due to that we are more familiar with the literatures from statistics. Since we are afraid to mis-state the essential results from those statistical physics ( or spin glasses) literatures, we only cited the thesis of Jean Barbier which we have read carefully and suggested the interested readers to learn more from its references. We agree with you that we should read through most of the related literatures and $\textbf{cited them explicitly}$. We have updated these references in the new draft.
>
> We totally understand that it might be trivial for expert as you to trust the equivalence between AMP state equation and cavity method based on some `folklore' in the spin glasses community. We are really glad to know if there is a published (mathematically) $\textbf{rigorous proof}$ of the equivalence.

---

> ### Comment · Reviewer_rXB2 · 2021-11-24
> **Acknowldging the authors feedback**
>
> I thanks the authors for their feedback. But I will stick to my initial decision.

---

### Official Review · Reviewer_U2rr · 2021-11-05

**Correctness:** 4
**Technical Novelty And Significance:** 2
**Empirical Novelty And Significance:** Not applicable
**Recommendation:** 3
**Confidence:** 4

**Main Review:**

-- Strengths:
- Establishing the connection between these different state evolutions is interesting. Of course we expected them to all have the same fixed points for MSE but showing that even the recursions are the same is novel and interesting.

-- Weaknesses:
- The main issue I have with this paper is significance and novelty of the results. As I said, it is interesting to show that the recursion can be turned into each other for all these methods, but the paper does not give any intuition what the implications of such results are? The authors just say that they hope this leads to finding a deeper structure in such high-dimensional problems. I find it interesting to see that the SEs can be turned into each other, but I believe it is not surprising and unless showing this equivalence leads to more interesting results, I do not see this result to be significant enough to be accepted for ICLR.

- Theorem 3 is not new. It is well known that fixed points of AMP correspond to critical points of Lasso objective function and the min-max problem in CGMT is of course equivalent to Lasso which is essentially the result of the theorem. In fact, AMP and its variants such as VAMP can be used to solve many problems and showing that their fixed points if they converge satisfies the first order optimality conditions is rather easy.

-- Minor comments/typos:
- The paper has some grammatical and typographical errors. It needs a quick revision to correct for such errors. For example:
1. spacing between a word and parenthesis after it is often missing.
2.  beginning of page 8: self-content
3. remark 3.1: equivalent -> equivalence
and many more.

**Summary Of The Paper:**

The paper establishes equivalence of state evolution for approximate message passing (AMP), convex min-max Gaussian theorem (CGMT), and Leaving-one-out (LOO) for three different problems: 1. a robust linear model estimator called the M-estimator throughout the paper; 2. Lasso; and 3. logistic regression.

**Summary Of The Review:**

I believe the contributions of this paper are not significant enough.

---

> ### Author Response · Authors · 2021-11-21
> **Response to Reviewer U2rr**
>
> Thanks for your summary and detailed comments.  They show that you are a responsible reviewer and have great achievement in this field. I would like to clarify your concerns in the following:
>
> a): The main issue I have with this paper is significance and novelty of the results:
>
> Thank you for your suggestions. Indeed, the results in this paper is not shocking. Meanwhile, last version of my paper did not clarify the significance and novelty clearly in the introduction part.  In view of this drawback, we have $\textbf{revised the introduction}$ to emphasize the significance and novelty.
>
> $\textbf{About novelty}$:
>
> To the best of our knowledge, this is $\textbf{the first work}$ to $\textbf{clearly clarify}$ the equivalence among SEs derived from different methods and try to establish the equivalence of different methods.
>
> $\textbf{About significance: }$
>
> Three methods exhibit different reduction forms of original
> optimization. In specific problem, these methods have different strength
> and weakness. We want to find the inner relation about these methods. By
> revealing the equivalence of these methods, we want to provide a more
> straightforward method to derive SEs. Specifically, in view of Theorem3,
> we want to construct a method which can derive SEs from the first-order
> optimality condition directly.
>
> Establishing equivalence of SEs is the first step. The main purpose of
> this proposal is to communicate with other scholars and share our
> interesting observations. Deeper equivalence, such as equivalence of
> different methods, is hard to clarify at now since the absence of
> related work, as have clarified by reviewer e98M. We will establish
> related tools for establishing equivalence and try our best to clarify
> the deeper equivalence in the following work.
>
> However, we emphasize that even the equivalence of SEs has significant
> influence, as is shown in the following.
>
> b): The paper does not give any intuition what the
> implications of such results are. I believe it is not surprising and
> unless showing this equivalence leads to more interesting
> results.
>
>
> Thank you for your comments. Equivalence of SEs maybe not too very
> surprising. But we think it is meaningful and significant:
>
> 1-The equivalence of SEs is $\textbf{the most direct, accessible, basic way}$ to
> establish the equivalence. And it is the $\textbf{basis}$ of more insightful
> equivalence.
>
> 2-Since parameters in different methods express different statistical
> meanings in their own ways. The parameter transformations can help us
> $\textbf{related the statistical meanings in different methods}$, which can help us
> establish the equivalence of three methods. The strong similarity of
> parameter transformations in different settings(say, different theorems)
> can inspire us to explore equivalence of three methods dedicatedly,
> which is our further goal.
>
> 3-For the equivalence of SEs itself, it can serve as a way for
> researchers using one method to $\textbf{check their predictions with another
> method using a parameter transformation}$ suggested in this paper, which
> has indicated in the comment of reviewer e98M.
>
> We will try our best to illustrate further interesting results in the
> following work.
>
> c): Showing that their fixed points if they converge satisfies
> the first order optimality conditions is rather
> easy.
>
> Indeed, this result is easy and can be predictable. This result is shown
> to clarify that both AMP and CGMT are derived SEs from the first-order
> optimality condition so they have some inner equivalence to some extent.
>
> Meanwhile, we have revised Theorem 3 as a proposition in the new
> version.
>
> $\textbf{typo:}$
>
>
> Thank you for pointing out our typos patiently and we have corrected
> this in the revised paper.

---

### Author Response · Authors · 2021-11-21
**General Comment**

Thanks for your valuable suggestions. They are very important for us to improve the manuscript.

First, We would like to provide a sketch of our original purpose of this project and hope that it may help the readers understand the significance and novelty of this piece of note.

For certain high dimensional statistical problem (e.g., Lasso-estimator and the M-estimator), several methods such as AMP, LOO and CGMT are proposed to obtain the exact risk (not merely the rate).  Since these methods are solving the same problem, all of them should  share some characteristics in principle.

For example, the risk are characterized by different set of 'state equations' in different methods. It is trivial that all the three sets of equations must possess the same solution. A slightly non-trivial question is that if there are some canonical way to establish the equivalence between these three sets of equations. The ultimate question for us is looking for a conceptual/rigorous understanding between the three approaches and we hope that this new understanding can bring us more tools in dealing with high dimensional statistical problems.

The current manuscript illustrates that the equivalence is somehow `canonical' which in turn strongly suggests that there should be a conceptual/rigorous understanding between the three approaches.  We have to emphasize that though many papers have explicitly written down the corresponding state equations, none of them have shown that these sets of state equations are compatible. To the best of our knowledge, only [1] mentioned  there is another set of state equation (without any comparison).

We recognize that our solution is somehow limited. We submit this manuscript mainly because that we believe that the underlying reason of the equivalence appeared here might be an interesting question and we are more than happy to see its solution from other researchers.





$\textbf{Reference}$:
[1]Deng Z, Kammoun A, Thrampoulidis C. A model of double descent for high-dimensional binary linear classification[J]. arXiv preprint arXiv:1911.05822, 2019.


The responses to the reviewer's comments and detail corrections are as follows:

---

### Decision · Program_Chairs · 2022-01-20

**Decision:**

Reject

**Comment:**

Three out of the four reviews rated this paper well below the acceptance threshold. Although the review scores show a relatively large spread, I think that the review contents are more or less coherent across the four reviewers.
The equivalence of the state equations (SEs; a set of equations that macroscopically characterizes optimal solutions of certain high-dimensional regression problems) derived from three different approaches (AMP, CGMT, and LOO) is well expected to hold, as the optimal solutions should be independent of how their macroscopic characterization in the form of an SE is derived, and this paper concretely showed such equivalence to hold for three problems. More concretely, Theorem 1 states the equivalence of the SEs for M-estimator derived from the three approaches, Theorem 2 states the equivalence of the SEs for LASSO derived from AMP and CGMT, and Theorem 4 states the equivalence of the SEs for logistic regression derived from LOO and CGMT. The main concern raised by all the reviewers is that this paper does not provide novel and significant insights as to why and how the equivalence arises.
Some reviewers also pointed out that this paper lacks citation to the relevant statistical-mechanics literature, as well as that this paper contains so many typos, grammatical errors, and inappropriate typesetting styles. The authors responses were not instrumental in persuading the reviewers with negative evaluation. On the basis of these I would not be able to recommend acceptance of this paper for presentation at ICLR 2021.